# STITCH-OPE: Trajectory Stitching with Guided Diffusion for Off-Policy Evaluation

**Hossein Goli**[1,2,4]    **Michael Gimelfarb**[1,2,4]    **Nathan De Lara**[1,2,4]    **Haruki Nishimura**[3]
**Masha Itkina**[3]    **Florian Shkurti**[1,2,4]
[1]Department of Computer Science, University of Toronto
[2]University of Toronto Robotics Institute, Toronto, Canada
[3]Toyota Research Institute, Los Altos, California
[4]Vector Institute, Toronto, Canada
hossein.goli@mail.utoronto.ca    mike.gimelfarb@mail.utoronto.ca
nathan.delara@mail.utoronto.ca    haruki.nishimura@tri.global
masha.itkina@tri.global    florian@cs.toronto.edu

## Abstract

Off-policy evaluation (OPE) estimates the performance of a target policy using offline data collected from a behavior policy, and is crucial in domains such as robotics or healthcare where direct interaction with the environment is costly or unsafe. Existing OPE methods are ineffective for high-dimensional, long-horizon problems, due to exponential blow-ups in variance from importance weighting or compounding errors from learned dynamics models. To address these challenges, we propose STITCH-OPE, a model-based generative framework that leverages denoising diffusion for long-horizon OPE in high-dimensional state and action spaces. Starting with a diffusion model pre-trained on the behavior data, STITCH-OPE generates synthetic trajectories from the target policy by guiding the denoising process using the score function of the target policy. STITCH-OPE proposes two technical innovations that make it advantageous for OPE: (1) prevents over-regularization by subtracting the score of the behavior policy during guidance, and (2) generates long-horizon trajectories by stitching partial trajectories together end-to-end. We provide a theoretical guarantee that under mild assumptions, these modifications result in an exponential reduction in variance versus long-horizon trajectory diffusion. Experiments on the D4RL and OpenAI Gym benchmarks show substantial improvement in mean squared error, correlation, and regret metrics compared to state-of-the-art OPE methods [†]

## 1 Introduction

Given the slow and risky nature of online data collection, real-world applications of reinforcement learning often require offline data for policy learning and evaluation [23, 39]. An important problem of working with offline data is *off-policy evaluation* (OPE), which aims to evaluate the performance of target policies using offline data collected from other behavior policies. One practical advantage of OPE is that it saves the cost of evaluation on hardware in embodied applications in the real world [28]. However, a central challenge of OPE is the presence of *distribution shift* induced by differences in behavior and target policies [23, 3]. This can lead to inaccurate estimates of policy values, making it difficult to trust or select between multiple target policies before they are deployed [44, 26].

Numerous approaches have attempted to address the distribution shift in offline policy evaluation by reducing either the variance of the policy value or its bias, but they are typically ineffective in

---

[†]Project website and code: **stitch-ope.github.io**.

39th Conference on Neural Information Processing Systems (NeurIPS 2025).

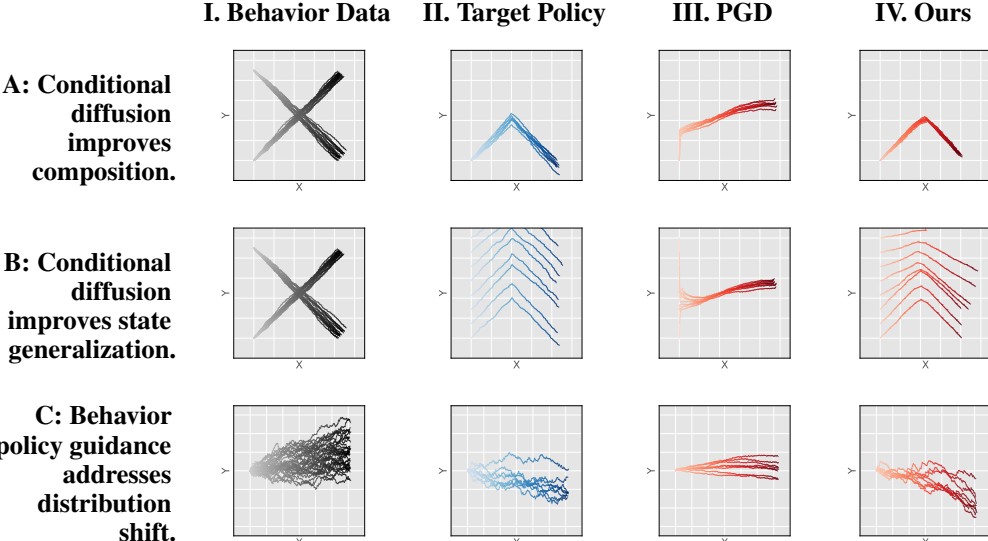

Table 1: A 2D toy problem with Gaussian dynamics illustrates the advantages of STITCH-OPE. **Row A:** Behavior data is a mixture of two datasets generated by different behavior policies $\beta_1$ and $\beta_2$. The target policy in column II is a piecewise function following $\beta_1$ on the left half-space and $\beta_2$ on the right half-space. Policy-Guided Diffusion (PGD) in column III estimates the behavior trajectory distribution by training a diffusion model on the behavior dataset, then leverages guided diffusion to generate target policy trajectories [16]. PGD is unable to stitch behavior trajectories correctly while STITCH-OPE is. One explanation is that conditional diffusion provides a higher entropy sampling distribution than full-length diffusion, and thus ensures a broader coverage of the modes in the behavior data (see Section 3.3 for details). **Row B:** The initial state is varied during trajectory generation. As shown in column III, PGD cannot generalize to new initial states. On the contrary, STITCH-OPE is trained on sub-trajectories that start in arbitrary states in the behavior dataset as opposed to only initial states, which allows it to achieve better generalization. **Row C:** A scenario with severe distribution shift is presented, where the behavior and target policies move the agent in different directions. As shown in columns III and IV, the negative behavior guidance term is essential to prevent over-regularization, which can prevent guided diffusion from addressing distribution shift and lead to biased value estimation.

high-dimensional long-horizon problems. For example, Importance Sampling (IS) [33] estimates the value of the target policy by weighing the behavior policy rollouts according to the ratio of their likelihoods. However, it suffers from the so-called *curse of horizon* where the variance of the estimate increases exponentially in the evaluation horizon [25]. More recent model-free OPE estimators reduce or eliminate the explosion in variance by estimating the long-run state-action density ratio $d^\pi(s, a)/d^\beta(s, a)$ between the target and behavior policy [25, 31, 43], yet they have demonstrated poor empirical performance on high-dimensional tasks where the behavior and target policies are different (i.e. the behavior policy is not a noisy version of the target policy) [14].

As an alternative approach, model-based OPE estimators typically learn an empirical autoregressive model of the environment and reward function from the behavior data, which is used to generate synthetic rollouts from the target policy for offline evaluation [19, 38, 46]. Some advantages of the model-based paradigm include sample efficiency [24], exploitation of prior knowledge about the dynamics [12], and better generalization to unseen states [45]. Although model-based OPE methods often scale well to high-dimensional short-horizon problems – owing to the scalability of the deep model-based RL paradigm – their robustness diminishes in long-horizon tasks due to the compounding of errors in the approximated dynamics model [14, 17, 18].

Driven by the recent successes of generative diffusion in RL [29, 48, 1, 16, 30, 35], we propose ***Sub-Trajectory Importance-Weighted Trajectory Composition for Long-Horizon OPE*** for model-based off-policy evaluation in long-horizon high-dimensional problems. STITCH-OPE first trains a diffusion model on behavior data, allowing it to generate dynamically feasible behavior trajectories [18].

STITCH-OPE differs from prior work by training the diffusion model on short sub-trajectories instead of full rollouts, where sub-trajectory generation is conditioned on the final state of the previous generated sub-trajectory. This enables accurate trajectory "stitching" using short-horizon rollouts, while minimizing compounding error of full-trajectory rollouts, thus bridging the gap between model-based OPE and full-trajectory offline diffusion.

STITCH-OPE explicitly accounts for distribution shift in OPE by guiding the diffusion denoising process [9, 18] during inference. This can be achieved by selecting the guidance function to be the difference between the score functions of the target and behavior policies. A significant advantage of guided diffusion is that it eliminates the need to retrain the diffusion model for each new target policy. By pretraining the model on a variety of behavior datasets, generalization can be achieved during guided sampling to produce feasible trajectories under the target policy, leading to robust off-policy estimates for target policies that lack offline data. STITCH-OPE contributes the following novel technical innovations that we consider critical to the successful and robust application of diffusion models for OPE:

- **Trajectory Stitching with Conditional Diffusion.** We propose a state-conditioned guided diffusion model for generating short sub-trajectories from the target policy (Figure 1). This significantly improves the quality (Table 1, row A) and generalization (Table 1, row B) of trajectory generation in long-horizon tasks. Theorem 3.3 also provides theoretical bounds on the bias and variance of our proposed approach.

- **Behavior Guidance.** We show that the negative score function of the behavior policy mitigates the diffusion model from collapsing to trajectories with large behavior likelihood, thus improving generalization out-of-data (Table 1, row C). This negative score function naturally arises from viewing the likelihood ratio (i.e. in importance sampling) as the classification density in diffusion guidance [9] – a connection missed in prior work [16].

- **Robustness Across Problem Difficulty.** Finally, we evaluate STITCH-OPE on the OpenAI Gym control suite [4] and the D4RL offline RL suite [13], showing significant improvements compared to other recent OPE estimators across a variety of metrics (mean squared error, rank correlation and regret), problem dimension and evaluation horizon. To our knowledge, STITCH-OPE is the first work to demonstrate robust off-policy evaluation in high-dimensional long-horizon tasks.

## 2 Preliminaries

**Markov Decision Processes.** We consider the standard *Markov Decision Process* (MDP), which consists of a 6-tuple $\langle \mathcal{S}, \mathcal{A}, R, P, d_0, \gamma \rangle$ [34] where: $\mathcal{S}$ is the continuous state space, $\mathcal{A}$ is the continuous action space, $R$ is the reward function, $P$ is the Markov transition probability distribution, $d_0$ is the initial state distribution and $\gamma \in [0, 1]$ is the discount factor.

A policy $\pi : \mathcal{S} \times \mathcal{A} \to [0, \infty)$ observes the current state $s$, samples an action according to its conditional distribution $\pi(\cdot|s)$, and observes the immediate reward $R(s, a)$ and the next state $s' \sim P(\cdot|s, a)$. The interaction of a policy with an MDP generates a set of trajectories of states, actions, and rewards. The goal of reinforcement learning is to learn a policy $\pi$ that maximizes the expected return:

$$J(\pi) = \mathbb{E}_{\tau \sim p_\pi} \left[ \sum_{t=0}^{T-1} \gamma^t R(s_t, a_t) \right], \qquad p_\pi(\tau) = d_0(s_0) \prod_{t=0}^{T-1} \pi(a_t|s_t) P(s_{t+1}|s_t, a_t)$$

over length-$T$ trajectories $\tau = (s_0, a_0, s_1, a_1, \ldots s_T)$ induced by policy $\pi$.

**Off-Policy Evaluation.** The goal of *Off-Policy Evaluation* (OPE) is to estimate the expected return of some *target policy* $\pi$ given only a data set of trajectories $\mathcal{D}_\beta$ from some *behavior policy* $\beta$. To estimate $J(\pi)$, it is necessary to approximate the distribution over trajectories $p_\pi(\tau)$ induced by $\pi$ using only samples from the distribution $p_\beta(\tau)$. Hence, the phenomenon of *distribution shift* arises whenever $\beta$ is sufficiently different from $\pi$, in which case $p_\beta(\tau)$ is not a suitable proxy for obtaining samples from $p_\pi(\tau)$ and corrections must be made to account for the distribution shift.

**Denoising Diffusion Models.** *Denoising Diffusion Probabilistic Models* (DDPMs) [36, 15] are a class of generative models to sample from a given distribution. Given a dataset $\{x_i\}_{i=1}^N$ and the

$K$-step (forward) noise process $x_i^k = \sqrt{\alpha_k} x_i^{k-1} + \sqrt{1-\alpha_k}\epsilon$, where $\epsilon \sim \mathcal{N}(0, I)$ is independent, DDPMs are trained to perform the $K$-step (backward) denoising process to recover $x_i$ from $x_i^K \sim \mathcal{N}(0, I)$. This is accomplished by running the forward process $x_i^0 \to x_i^k$ on the original $x_i$ and training the DDPM $\epsilon_\theta$ on the denoising process $x_i^{k-1}|x_i^k \sim \mathcal{N}(\mu(x_i^k, k), \sigma_k^2 I)$ to predict the noise $\epsilon$ so that the distribution over denoised samples $x_i^0$ matches $x_i$. The standard reparameterization $\mu(x_i^k, k) = (x_i^k - \epsilon_\theta(x_i^k, k)(1-\alpha_k)/\sigma_k)/\sqrt{\alpha_k}$ maps $x^k$ and the predicted noise $\epsilon_\theta(x_i^k, k)$ to $x^{k-1}$ to undo the noise accumulated during the forward process. The following loss function is typically used to train a diffusion model [15]

$$\mathcal{L}(\theta) = \mathbb{E}_{k, x_i, \epsilon \sim \mathcal{N}(0, I)} \left[ \|\epsilon - \epsilon_\theta(x_i^k, k)\|^2 \right]. \tag{1}$$

**Guided Diffusion.** It is possible to guide the sampling from a trained diffusion model to maximize some classifier $p(y|x)$ [9]. A key observation of diffusion is that the backward diffusion process can be well approximated by a Gaussian when the noise is small, that is, $x^k|x^{k+1} \sim \mathcal{N}(\mu_k, \Sigma_k)$. Next, observe that $p(x^k|x^{k+1}, y) \propto p(x^k|x^{k+1})p(y|x^k)$ by Bayes' rule. Applying the first-order Taylor approximation $\log p(y|x^k) \approx \log p(y|\mu_k) + (x^k - \mu_k)g(\mu_k)$ at $\mu_k$, where $g(u) = \nabla_x \log p(y|x)|_{x=u}$, it can be shown that:

$$\log(p(x^k|x^{k+1})p(y|x^k)) \propto -\frac{1}{2}(x^k - \mu_k - \Sigma_k g(\mu_k))^T \Sigma_k^{-1}(x^k - \mu_k - \Sigma_k g(\mu_k))$$
$$\propto \log \mathcal{N}(x^k; \mu_k + \Sigma_k g(\mu_k), \Sigma_k). \tag{2}$$

In other words, it is possible to sample from the conditional (guided) distribution $p(x^k|x^{k+1}, y)$ by sampling from the original diffusion model with its mean shifted to $\mu_k + \Sigma_k g(\mu_k)$. Appendix A provides a worked example illustrating guided diffusion for a mixture of Gaussians.

## 3 Proposed Methodology

The *direct method* for off-policy evaluation [11] estimates the single-step autoregressive model $\hat{P}(s_t|s_{t-1}, a_{t-1})$ and the reward function $\hat{R}(s_t, a_t)$ from the behavior data. Then, it draws target policy trajectories $\tau \sim p_\pi(\tau)$ by forward sampling, that is, $s_0 \sim d_0, a_0 \sim \pi(\cdot|s_0), s_1 \sim \hat{P}(\cdot|s_0, a_0), \ldots s_T \sim \hat{P}(\cdot|s_{T-1}, a_{T-1})$. However, even small errors in $\hat{P}$ can lead to significant bias in $J(\pi)$ due to the compounding of errors over long horizon $T$ [20, 17]. STITCH-OPE avoids the compounding problem by generating the partial trajectory in a single (backward diffusion) pass, leading to more accurate OPE estimates over a long horizon.

### 3.1 Guided Diffusion for Off-Policy Evaluation

It is possible to approximate $p_\pi$ using guided diffusion by interpreting each data point $x_i$ as a full trajectory $\tau$. Given a behavior policy $\beta$ and corresponding length-$T$ trajectory distribution $p_\beta(\tau)$, the corresponding length-$T$ trajectory distribution of target policy $\pi$ can be written as:

$$p_\pi(\tau) = d_0(s_0) \prod_{t=0}^{T-1} \beta(a_t|s_t) P(s_{t+1}|s_t, a_t) \frac{\pi(a_t|s_t)}{\beta(a_t|s_t)} = p_\beta(\tau) \prod_{t=0}^{T-1} \frac{\pi(a_t|s_t)}{\beta(a_t|s_t)}, \tag{3}$$

which is the standard importance sampling correction [33]. We address the question of tractably learning $p_\beta(\tau)$ by training a diffusion model $\hat{p}_\beta(\tau)$ on the offline behavior data set $\mathcal{D}_\beta$ [18], thus approximating $\hat{p}_\beta(\tau) \approx p_\beta(\tau)$. Specifically, the diffusion model learns to map a trajectory consisting of pure noise, $\tau^k = (s_0^k, a_0^k, \ldots s_T^k)$, to a noiseless behavior trajectory $\tau^0 = (s_0^0, a_0^0, \ldots s_T^0)$.

A key observation is that we can bypass importance sampling in (3) by guiding the generation process $\hat{p}_\beta(\tau)$ towards $p_\pi(\tau)$ using diffusion guidance (2) [18]. Specifically, let $x^k = \tau^k$ denote a noisy behavior trajectory at step $k$ of the forward diffusion process, and let $y \in \{0, 1\}$ be a binary outcome with $p(y = 1|\tau) \propto \prod_{t=0}^{T-1} \frac{\pi(a_t|s_t)}{\beta(a_t|s_t)}$. Intuitively, $y$ indicates whether the trajectory $\tau$ is generated by the target policy $\pi$ ($y = 1$) or the behavior policy $\beta$ ($y = 0$), and the likelihood ratio determines the odds that $y = 1$ given $\tau$. By (3),

$$p_\pi(\tau) \propto p_\beta(\tau)p(y = 1|\tau),$$

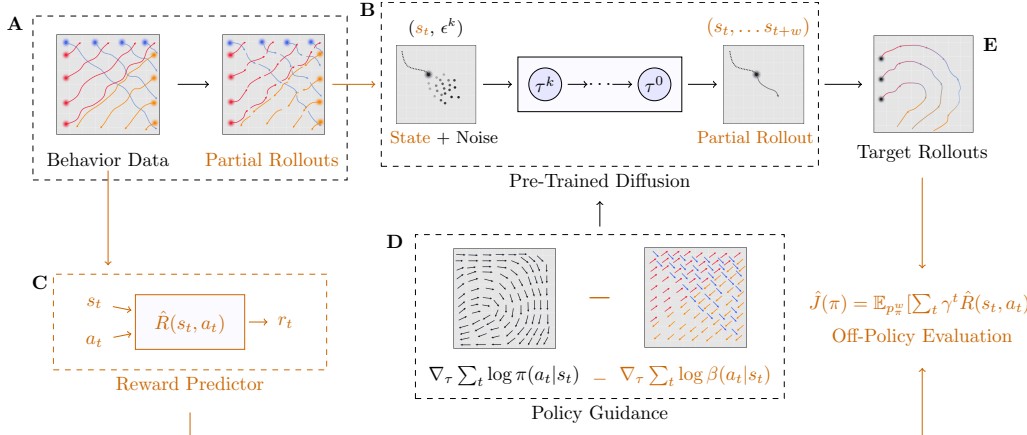

Figure 1: A conceptual illustration of STITCH-OPE, with novel contributions highlighted in orange. **A:** Behavior data is sliced into partial trajectories of length $w$. **B:** The data is fed to a conditional diffusion model taking a $w$-length sequence of Gaussian noise $\epsilon$ and state $s_t$ as inputs, and applies the backward diffusion process to predict the behavior trajectory of length $w$ beginning in state $s_t$. **C:** To evaluate policies, STITCH-OPE also trains a neural network on the behavior transitions to predict the immediate reward. **D:** It then applies guided diffusion on the pretrained diffusion model to generate a batch of partial target trajectories of length $w$, where the guidance function incorporates the score function of the target policy and the behavior policy. **E**: The guided partial trajectories are stitched end-to-end to produce full-length target trajectories. Finally, the guided trajectories are evaluated using the empirical reward function $\hat{R}(s, a)$, and averaged to estimate the value of the target policy.

and thus the backward diffusion process for generating target policy trajectories for OPE can be approximated with guidance (2):

$$
\begin{aligned}
\log p_\pi(\tau^k|\tau^{k+1}) &\propto \log(p_\beta(\tau^k|\tau^{k+1})p(y=1|\tau^{k+1})) \\
&\approx \log \mathcal{N}(\tau^{k+1}; \mu_k + \Sigma_k \nabla_\tau \log p(y=1|\tau)|_{\tau^{k+1}}, \Sigma_k),
\end{aligned}
\tag{4}
$$

where $p_\beta(\tau^k|\tau^{k+1}) = \mathcal{N}(\mu_k, \Sigma_k)$ is the backward diffusion process. Therefore, we can obtain feasible target policy trajectories using the guidance function:

$$
g(\tau) = \nabla_\tau \log p(y=1|\tau) = \nabla_\tau \sum_{t=0}^{T-1} \log \pi(a_t|s_t) - \nabla_\tau \sum_{t=0}^{T-1} \log \beta(a_t|s_t).
\tag{5}
$$

Given the approximate sampling distribution over the trajectories of the target policy described above, $\hat{p}_\pi(\tau^0) = \int \cdots \int \mathcal{N}(\tau^K; 0, I) \prod_{k=1}^{K} p_\pi(\tau^{k-1}|\tau^k) \, d\tau^K \ldots d\tau^1$, and an empirical reward function $\hat{R}(s, a)$, it is straightforward to estimate the expected return (or a statistic such as variance or quantile) given *any* target policy, i.e. $\hat{J}(\pi) = \mathbb{E}_{\tau \sim \hat{p}_\pi} \left[ \sum_t \gamma^t \hat{R}(s_t, a_t) \right] \approx J(\pi)$.

## 3.2 Negative Behavior Guidance

The target policy score function, $g_{simple}(\tau) = \nabla_\tau \sum_{t=0}^{T-1} \log \pi(a_t|s_t)$ [16], provides a simple guidance function for OPE. However, it corresponds to a biased estimator of $p_\pi(\tau)$ in the context of (3) and can generate trajectories that are unlikely under the target policy, as illustrated using the GaussianWorld domain in Table 1 (see Appendix B for details). The behavior policy $\beta$ returns a positive angle in each state and the target policy $\pi$ returns a negative angle, to test the performance of both guidance functions under distribution shift. Conclusions are summarized in row C of Table 1. The omission of the negative guidance term results in a sampling distribution that collapses to a high-density region under $p_\beta(\tau)$, where $p_\pi(\tau)$ could be small. In other words, **behavior guidance prevents the guided sampling distribution $\hat{p}_\pi$ from becoming over-regularized.**

In our empirical evaluation, we employ the following generalization of (5) to allow fine-grained control over the relative importance of the target and behavior policy guidance

$$g(\tau) = \alpha \nabla_\tau \sum_{t=0}^{T-1} \log \pi(a_t|s_t) - \lambda \nabla_\tau \sum_{t=0}^{T-1} \log \beta(a_t|s_t). \tag{6}$$

Ignoring the normalizing constant which does not dependent on $\tau$, (6) is equivalent to sampling from the following re-weighted trajectory distribution

$$q_\pi(\tau) \propto p_\beta(\tau) \prod_{t=0}^{T-1} \frac{\pi(a_t|s_t)^\alpha}{\beta(a_t|s_t)^\lambda}, \tag{7}$$

which can be interpreted as a *tempered posterior distribution* [2, 5] over trajectories; the importance of the likelihood terms associated with $\pi$ and $\beta$ are controlled by the choice of $\alpha$ and $\lambda$, respectively. Note that the choice $\alpha = \lambda = 1$ reduces to the standard guidance function $g(\tau)$. This is not a good empirical choice because it can push the backward diffusion process too far from the behavior distribution, leading to infeasible trajectories or instability. Instead, typical choices satisfy $\lambda < \alpha$ (see Appendix L.1 for an additional experiment confirming this). The choice $\alpha = 1, \lambda = 0$ reduces to $g_{simple}(\tau)$ and is unsuitable for OPE.

### 3.3 Sub-Trajectory Stitching with Conditional Diffusion

Recent work has shown that full-length diffusion models do not provide sufficient compositionality for accurate long-horizon sequence generation [6]. In addition, full-length prediction requires the generation of sequences of length $T \cdot (\dim(\mathcal{A}) + \dim(\mathcal{S}))$; this may be infeasible or inefficient on resource-constrained systems, when $T$ is large or when $\mathcal{A}$ or $\mathcal{S}$ is high-dimensional.

To tackle these limitations, STITCH-OPE trains a conditional diffusion model to generate behavior sub-trajectories of length $w \ll T$. To allow for a more flexible composition of behavior trajectories during guidance, generation in STITCH-OPE is performed in a semi-autoregressive manner from the diffusion model, which is conditioned on the last state of the previously generated sub-trajectory.

Specifically, the conditional diffusion model in STITCH-OPE, denoted as $\epsilon_\theta(\tau_{t:t+w}^k, k|s_t^0)$, denoises a length-$w$ noisy sub-trajectory $\tau_{t:t+w}^k = (s_t^k, a_t^k, s_{t+1}^k, a_{t+1}^k, \ldots s_{t+w-1}^k, a_{t+w-1}^k, s_{t+w}^k)$ conditioned on the last state $s_t^0$ of the previously generated sub-trajectory $\tau_{t-w:t}^0$. Generalizing (1), the loss function of STITCH-OPE is thus

$$\mathcal{L}_{STITCH-OPE}(\theta) = \mathbb{E}_{k,t,\tau_{t:t+w}\sim\mathcal{D}_\beta,\epsilon\sim\mathcal{N}(0,I)} \left[\|\epsilon - \epsilon_\theta(\tau_{t:t+w}^k, k|s_t^0)\|^2\right].$$

Next, writing $p_\beta(\tau_{t:t+w}|s_t^0)$ to denote the sampling distribution over fully denoised sub-trajectories $\tau_{t:t+w}^0$ conditioned on $s_t^0$, the sampling process (3) of STITCH-OPE can be written as:

$$p_\pi^w(\tau) = \prod_{t=0}^{T/w-1} \left(p_\beta(\tau_{wt:w(t+1)}|s_{wt}^0) \cdot \prod_{u=wt}^{w(t+1)-1} \frac{\pi(a_u|s_u)}{\beta(a_u|s_u)}\right), \tag{8}$$

where target trajectories are generated by guiding the conditional diffusion model (analogous to (4)) according to

$$g(\tau_{wt:w(t+1)}) = \alpha \nabla_{\tau_{wt:w(t+1)}} \sum_{u=wt}^{w(t+1)-1} \log \pi(a_u|s_u) - \lambda \nabla_{\tau_{wt:w(t+1)}} \sum_{u=wt}^{w(t+1)-1} \log \beta(a_u|s_u).$$

A complete algorithm description of STITCH-OPE is provided in Appendix E.

To understand the intuition that the conditional diffusion model offers better compositionality than the full-horizon prediction, we decompose the behavior trajectory distribution as a mixture over the trajectories $\tau_j$ in $\mathcal{D}_\beta$:

$$p_\beta(s_t, a_t, \ldots s_{T-1}|s_0, a_0, \ldots s_t) \approx \sum_{\tau_j \in \mathcal{D}_\beta} p_\beta(s_t, a_t, \ldots s_{T-1}|s_t, \tau_j)p(\tau_j|s_0, a_0, \ldots s_t).$$

Meanwhile, the conditional diffusion model ignores the full history of past states, i.e.:

$$p_\beta(s_t, a_t, \ldots s_{T-1}|s_0, a_0, \ldots s_t) \approx \sum_{\tau_j \in \mathcal{D}_\beta} p_\beta(s_t, a_t, \ldots s_{T-1}|s_t, \tau_j)p(\tau_j|s_t).$$

$p(\tau_j|s_t)$ has higher entropy than $p(\tau_j|s_0, a_0, \ldots s_t)$ since it is conditioned on less information (see Appendix C for a proof), and thus provides a broader coverage of the diverse modes in the behavior dataset. This improves the compositionality of guided long-horizon trajectory generation. Row A of Table 1 illustrates this claim empirically using the GaussianWorld problem. A further claim is that the STITCH-OPE model can generalize better across initial states with low, or even zero, probability under $d_0$ (see row B of Table 1). This occurs because $p_\beta^w$ is trained on sub-trajectories starting in arbitrary states in $\mathcal{D}_\beta$, as opposed to states sampled only from $d_0$. Therefore, **sliding windows strike an optimal balance between autoregressive methods and full-length trajectory diffusion, providing good compositionality while avoiding the error compounding in terms of $T$.**

### 3.4 Theoretical Analysis

We provide theoretical guarantees for our proposed STITCH-OPE method by analyzing its bias and variance. The first prerequisite assumption is standard in OPE [42, 27] and limits the ratio $\pi/\beta$.

**Assumption 3.1.** There is a constant $\kappa$ such that $\frac{\pi(a|s)}{\beta(a|s)} \leq \kappa$ for all $s \in \mathcal{S}, a \in \mathcal{A}$.

The second prerequisite assumption bounds the total variation between the learned length-$w$ trajectory distribution $\hat{p}_\beta^w$ and the true distribution $p_\beta^w$ under the behavior policy $\beta$.

**Assumption 3.2.** $TV(p_\beta^w, \hat{p}_\beta^w) \leq \delta_\beta$ for some constant $\delta_\beta$.

Our main result is a bound on the mean squared error of the STITCH-OPE estimator. We defer the full proofs, technical lemmas, and definitions to Appendix D.

**Theorem 3.3.** *Define $\hat{p}_\pi$ as the (length-$T$) trajectory distribution of the guided diffusion model, and $p_\pi$ as the true trajectory distribution under the target policy $\pi$. Under Assumptions 3.1 and 3.2, the mean squared error (MSE) of the STITCH-OPE return $\hat{J}$ satisfies:*

$$\mathbb{E}_{\hat{p}_\pi}\left[(\hat{J} - J(\pi))^2\right] \leq \underbrace{\left(\frac{2B_w}{1-\gamma^w}\kappa^w\delta_\beta\right)^2}_{\text{Bias}^2} + \underbrace{10\left(\frac{T}{w}\right)^2 B_w^2\kappa^w\delta_\beta + \frac{8B_w^2}{1-\gamma^{2w}}\kappa^w\delta_\beta + \text{Var}_{p_\pi}(J)}_{\text{Variance}},$$

*where $B_w = \frac{1-\gamma^w}{1-\gamma}\sup_{s,a}|R(s,a)|$ is a bound on the maximum length-$w$ discounted return, and $J$ is the return under $p_\pi$.*

**Remarks.** Theorem 3.3 resembles the $O(\exp(cT))$ bound of IS-based methods [25, 27], but is expressed in terms of $w$ rather than $T$ (with a more favorable $O(T^2)$ dependence). Since $w$ is a fixed hyper-parameter typically chosen to be much smaller than $T$ in practice, **STITCH-OPE provides an exponential reduction in MSE versus both importance sampling and length-$T$ diffusion!** In Section 4, we validate this claim further by showing that STITCH-OPE outperforms full trajectory diffusion (PGD) on most benchmarks (see Appendix L.2 for an additional experiment confirming small $w > 1$ is ideal in practice). The MSE decreases as the error in the learned behavior model $\delta_\beta$ decreases. In practice, $\delta_\beta$ is easier to estimate and control than the error of the target density $p_\pi^w$. It is also important to note that the variance cannot be reduced below $\text{Var}_{p_\pi}(J)$, the intrinsic variance of the environment and the target policy.

## 4 Empirical Evaluation

Our empirical evaluation aims to answer the following research questions:

1. Does the combination of conditional diffusion and negative guidance (as hypothesized in Table 1) translate to robust OPE performance on standard benchmarks?
2. Is STITCH-OPE robust across problem size (e.g., state/action dimension, horizon)?
3. Is STITCH-OPE robust across different levels of optimality of the target policy and the classes of policies?

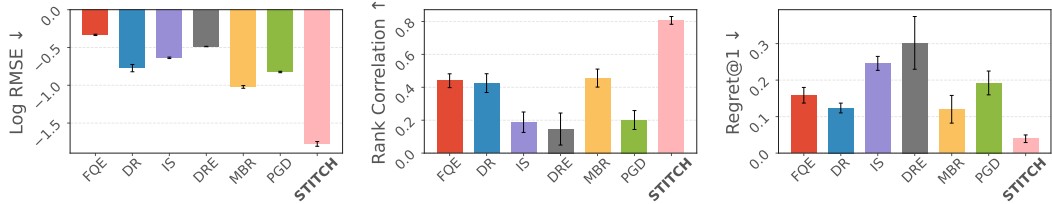

Figure 2: Mean overall performance of all baselines, averaged across environments. Error bars represent +/- one standard error.

|  | | FQE | DR | IS | DRE | MB | PGD | **Ours** |
|---|---|---|---|---|---|---|---|---|
| **Log RMSE →** | Hopper | -0.42 ± 0.03 | -0.57 ± 0.02 | -0.48 ± 0.01 | -0.42 ± 0.00 | -1.70 ± 0.04 | -1.22 ± 0.02 | **-2.33 ± 0.02** |
| | Walker2d | -0.48 ± 0.01 | -1.25 ± 0.08 | -0.71 ± 0.01 | -0.45 ± 0.00 | -0.88 ± 0.01 | -0.32 ± 0.01 | **-1.33 ± 0.01** |
| | HalfCheetah | -0.05 ± 0.00 | 0.01 ± 0.01 | -0.84 ± 0.02 | -1.19 ± 0.00 | -0.37 ± 0.00 | **-1.47 ± 0.00** | -0.85 ± 0.01 |
| | Pendulum | -0.58 ± 0.00 | -1.02 ± 0.04 | -0.15 ± 0.00 | -0.58 ± 0.00 | -0.43 ± 0.01 | -0.91 ± 0.01 | **-2.34 ± 0.07** |
| | Acrobot | -0.14 ± 0.00 | -0.49 ± 0.06 | -1.00 ± 0.01 | 0.20 ± 0.01 | -1.54 ± 0.02 | -0.13 ± 0.01 | **-2.02 ± 0.05** |
| **Rank Corr. ←** | Hopper | 0.17 ± 0.05 | 0.69 ± 0.06 | -0.06 ± 0.13 | -0.09 ± 0.14 | 0.52 ± 0.03 | 0.36 ± 0.09 | **0.76 ± 0.02** |
| | Walker2d | 0.41 ± 0.05 | 0.50 ± 0.02 | 0.51 ± 0.11 | 0.42 ± 0.05 | **0.65 ± 0.04** | -0.07 ± 0.10 | 0.63 ± 0.03 |
| | HalfCheetah | -0.03 ± 0.06 | -0.48 ± 0.07 | 0.57 ± 0.06 | 0.80 ± 0.02 | 0.32 ± 0.03 | 0.50 ± 0.01 | **0.87 ± 0.01** |
| | Pendulum | 0.89 ± 0.03 | 0.72 ± 0.07 | -0.60 ± 0.00 | -0.40 ± 0.15 | 0.84 ± 0.06 | 0.54 ± 0.02 | **0.96 ± 0.02** |
| | Acrobot | 0.75 ± 0.02 | 0.63 ± 0.08 | 0.52 ± 0.01 | 0.01 ± 0.12 | 0.53 ± 0.11 | 0.43 ± 0.14 | **0.82 ± 0.04** |
| **Regret@1 →** | Hopper | 0.34 ± 0.03 | 0.05 ± 0.02 | 0.13 ± 0.02 | 0.27 ± 0.17 | 0.04 ± 0.03 | **0.04 ± 0.01** | 0.11 ± 0.04 |
| | Walker2d | 0.23 ± 0.04 | 0.12 ± 0.00 | 0.09 ± 0.06 | 0.11 ± 0.00 | 0.05 ± 0.04 | 0.32 ± 0.16 | **<0.01 ± 0.00** |
| | HalfCheetah | 0.36 ± 0.00 | 0.37 ± 0.00 | 0.03 ± 0.01 | **<0.01 ± 0.00** | 0.32 ± 0.03 | 0.10 ± 0.00 | 0.08 ± 0.01 |
| | Pendulum | 0.03 ± 0.03 | 0.08 ± 0.03 | 0.98 ± 0.00 | 0.85 ± 0.13 | 0.07 ± 0.03 | 0.13 ± 0.00 | **<0.01 ± 0.01** |
| | Acrobot | 0.04 ± 0.01 | **<0.01 ± 0.00** | **<0.01 ± 0.00** | 0.28 ±0.06 | 0.10 ± 0.06 | 0.22 ± 0.06 | **0.01 ± 0.01** |

Table 2: Comparison of OPE methods across environments. Error bars represent ± one standard error across 5 seeds; any regret shown as <0.01 is nonzero but rounds to zero at two decimals.

## 4.1 Experiment Details

**Domains.** We evaluate the performance of STITCH-OPE in high-dimensional long-horizon tasks using the standard D4RL benchmark [13] and their respective benchmark policies [14]. Specifically, we use the `halfcheetah-medium`, `hopper-medium` and `walker2d-medium` behavior datasets. Each evaluation consists of 10 target policies $\pi_1, \pi_2, \ldots \pi_{10}$ trained at varying levels of ability [14]. We also carry out similar experiments using classical control tasks (Pendulum and Acrobot) from OpenAI Gym [4], to evaluate the competitiveness of STITCH-OPE on standard benchmarks on which other baselines have been extensively evaluated. For this set of environments, we obtain the target policies by running the twin-delayed DDPG algorithm [8] (see Appendix G for details). We set the trajectory length to $T = 768$ for all D4RL problems, $T = 256$ for Acrobot, and $T = 196$ for Pendulum. We also use $\gamma = 0.99$ in all experiments. The domain details are provided in Appendix F, and the training details of STITCH-OPE are provided in Appendix J.

**Baselines.** We include the following model-free estimators: **Fitted Q-Evaluation (FQE)** [22], **Doubly-Robust OPE (DR)** [37], **Importance Sampling (IS)** [33], and **Density Ratio Estimation (DRE)** [31]. We also include the following model-based estimators: **Model-Based (MB)** [19, 40], and **Policy-Guided Diffusion (PGD)** [16]. The implementation details are provided in Appendix H. Appendix L.6 provides an additional comparison against **Diffusion World Models (DWM)** [10].

**Metrics.** Each baseline method is evaluated on each pair of behavior dataset and target policy for 5 random seeds. We also generate ground-truth estimates of each target policy value by running each policy in the environment. We evaluate the performance of each baseline using the **Log Root Mean Squared Error (LogRMSE)**, the **Spearman Correlation**, and the **Regret@1** calculated as the difference in return between the best policy selected using the baseline policy value estimates and the actual best policy. Furthermore, to compare metrics consistently across tasks, we normalize the returns following [14]. Appendix I contains the technical details for metric calculation.

## 4.2 Discussion

Table 2 summarizes the performance of each method per domain, while Figure 2 summarizes the aggregated performance averaged across all domains. STITCH-OPE outperforms all baselines in 11 out of 15 instances (shown in bold), with general agreement among the different metrics. STITCH-OPE soundly outperforms both single-step (MB) and full-trajectory (PGD) model-based methods. This reaffirms our argument in Section 3.3 that intermediate values of $w$ provide a good balance between compositionality and compounding errors. Furthermore, STITCH-OPE performs particularly well according to rank correlation and Regret@1 (with very low standard error) and can accurately rank and identify the best-performing policy. This suggests that the target policy score function (with the negative behavior term) provides very informative guidance during denoising, allowing it to correctly evaluate target policies of varying levels of ability, even as some of those policies deviate significantly from the behavior policy. Finally, we see that STITCH-OPE performance remains consistent across the problem dimension, highlighting the scalability of diffusion when applied to OPE for high-dimensional problems.

## 4.3 Off-Policy Evaluation with Diffusion Policies

To demonstrate the ability of STITCH-OPE to evaluate more complex policy classes, we replace target policies with diffusion policies, which have led to significant advances in robotics [7, 41] (see Appendix K for details). Since STITCH-OPE only requires the score of the target policy, it is computationally straightforward to perform OPE with diffusion policies, which is not the case for other estimators that require an explicit probability distribution $\pi_i(a|s)$ over actions (i.e. IS, DR). D4RL results are provided in Table 3. We see that STITCH-OPE outperforms all other baselines in 6 out of 9 instances, demonstrating robust OPE performance across multiple target policy classes.

| | | FQE | DRE | MBR | PGD | **Ours** |
|---|---|---|---|---|---|---|
| Log RMSE ↓ | Hopper | -0.21 ± 0.01 | -0.38 ± 0.00 | -1.56 ± 0.02 | -0.89 ± 0.00 | **-1.65 ± 0.01** |
| | Walker2d | -0.59 ± 0.01 | -0.49 ± 0.00 | -0.81 ± 0.01 | -0.50 ± 0.00 | **-1.20 ± 0.01** |
| | HalfCheetah | -0.19 ± 0.00 | **-1.19 ± 0.00** | -0.24 ± 0.01 | -0.96 ± 0.00 | -0.50 ± 0.00 |
| Rank Corr. ↑ | Hopper | 0.35 ± 0.06 | 0.35 ± 0.04 | 0.68 ± 0.02 | 0.45 ± 0.00 | **0.81 ± 0.01** |
| | Walker2d | 0.03 ± 0.04 | 0.45 ± 0.03 | 0.47 ± 0.02 | **0.52 ± 0.01** | 0.46 ± 0.09 |
| | HalfCheetah | 0.59 ± 0.01 | 0.80 ± 0.03 | 0.75 ± 0.05 | 0.46 ± 0.06 | **0.81 ± 0.02** |
| Regret@1 ↓ | Hopper | 0.06 ± 0.03 | 0.41 ± 0.22 | 0.18 ± 0.00 | **<0.01 ± 0.00** | **<0.01 ± 0.00** |
| | Walker2d | 0.24 ± 0.02 | 0.59 ± 0.13 | 0.17 ± 0.02 | 0.23 ± 0.00 | **0.03 ± 0.00** |
| | HalfCheetah | **<0.01 ± 0.00** | **<0.01 ± 0.00** | 0.03 ± 0.01 | 0.02 ± 0.01 | 0.02 ± 0.01 |

Table 3: Comparison of OPE methods across environments when the target policy is a diffusion policy; any regret shown as <0.01 is nonzero but rounds to zero at two decimals.

## 4.4 Ablations

We conduct additional experiments to test the sensitivity of STITCH-OPE to the choice of guidance coefficients ($\alpha$ and $\lambda$), window size $w$, and data sets. Due to space limitations, we defer results to Appendix L. We summarize the key findings below:

- **Sensitivity to Penalty Coefficients $\alpha$ and $\lambda$ (Appendix L.1).** The best performance occurs when $0 < \lambda < \alpha$, reaffirming our claim in Section 3.2 that optimal regularization occurs for small $\lambda$.

- **Sensitivity to Window Size $w$ (Appendix L.2).** The best performance occurs for $w = 8$, showing that STITCH-OPE provides an optimal balance between autoregressive and full-trajectory diffusion.

- **Robustness Across Datasets (Appendix L.3).** We evaluated STITCH-OPE and all baseline methods on Hopper-medium-expert and Hopper-expert data sets. We found that STITCH-OPE outperforms all other baselines by a significant margin in 5 out of 6 instances (metric-dataset combinations) and maintained the most consistent performance across data sets.

| Horizon | FQE | DR | IS | MB | PGD | **Ours** |
|---------|-----|-----|-----|-----|-----|-----|
| $T = 128$ | -2.23 ± 0.00 | -3.49 ± 0.02 | -3.82 ± 0.01 | -3.66 ± 0.01 | -3.65 ± 0.02 | **-4.34 ± 0.01** |
| $T = 256$ | -1.36 ± 0.00 | -2.45 ± 0.00 | -2.04 ± 0.01 | -2.98 ± 0.01 | -3.03 ± 0.01 | **-3.19 ± 0.00** |
| $T = 512$ | -0.99 ± 0.00 | -1.65 ± 0.02 | -1.19 ± 0.01 | -2.17 ± 0.01 | -2.30 ± 0.01 | **-2.65 ± 0.01** |
| $T = 768$ | -0.81 ± 0.00 | -1.32 ± 0.01 | -0.94 ± 0.00 | -1.56 ± 0.01 | -1.82 ± 0.01 | **-1.99 ± 0.01** |
| $T = 1000$ | -0.69 ± 0.00 | -1.14 ± 0.02 | -0.81 ± 0.00 | -1.24 ± 0.01 | -1.45 ± 0.00 | **-1.61 ± 0.02** |

Table 4: Log RMSE performance on the Hopper-medium dataset across varied evaluation horizon $T$.

- **Sensitivity to Evaluation Horizon $T$.** Table 4 shows the log RMSE performance on the Hopper-medium data set for all baselines and various values of the evaluation horizon $T$. We also set $\gamma = 0.999$ to better see the performance differences between the methods for $T = 1000$. The values are normalized based on the global minimum and maximum across all evaluation horizon experiments. STITCH-OPE consistently outperforms all baselines for all values of $T$ considered, striking an optimal balance between auto-regressive and trajectory diffusion methods. Furthermore, STITCH-OPE performance slowly degrades as $T$ increases, validating the theoretical upper bound of Theorem 3.3.

- **Effect of Reward Estimation (Appendix L.4).** We compare STITCH-OPE performance given access to the true (unknown) reward function and its performance using reward estimation on Hopper-medium. The accumulated errors in the reward estimation have a minimal effect on the aggregate OPE performance. Furthermore, the error remains stable and uniformly low across all target policies.

## 5    Limitations

The theory of STITCH-OPE relies on (standard) Assumptions 3.1 and 3.2. In practice, if there exist many $(s, a)$ pairs such that $\beta(a|s) = 0$ but $\pi(a|s) > 0$, then the behavior data may be incomplete and diffusion guidance could generate infeasible trajectories and produce biased estimates of $J(\pi)$. Diffusion models trained on image data in other applications are often easy to interpret; however, evaluating the fidelity of the trajectories generated from the trained behavior model $p_\beta(\tau)$ is challenging when the state is complex and difficult to interpret or partially observable. Finally, while STITCH-OPE has demonstrated excellent performance on existing OPE benchmarks, it remains unanswered whether its benefits also apply to domain-specific problems outside robotics.

## 6    Conclusion

We presented STITCH-OPE for off-policy evaluation in high-dimensional, long-horizon environments. STITCH-OPE trains a conditional diffusion model to generate behavior sub-trajectories, and applies diffusion guidance using the score of the target policy to correct the distribution shift induced by the target policy. Our novelties include trajectory stitching and negative behavior policy guidance, which were shown to improve composition and generalization. Using D4RL and OpenAI Gym benchmarks, we showed that STITCH-OPE outperforms state-of-the-art OPE methods across MSE, correlation and regret metrics. Future work could investigate online data collection to address severe distribution shift, or explore ways to adapt the guidance coefficients or incorporate additional knowledge into the guidance function (e.g. additional structure on the dynamics). It also remains an open question whether the advantages of STITCH-OPE apply to offline policy optimization.

## Acknowledgments and Disclosure of Funding

We thank the anonymous reviewers and area chairs for their constructive feedback. This work was partially supported by the Toyota Research Institute (TRI).

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

# STITCH-OPE: Trajectory Stitching with Guided Diffusion for Off-Policy Evaluation

## Supplementary Material

## Abstract

This supplement to the paper discusses algorithmic and experiment details that were not included in the main paper due to space limitations. It includes proofs of all main theoretical claims, as well as all configurations and parameter settings that are required to reproduce the experiments. It includes additional experiments and ablation studies that were excluded from the main paper due to space limitations.

## Contents

# A  Pedagogical Example for Guided Diffusion

We consider the simple two-component mixture of Gaussians with density function

$$p(x) = 0.5\mathcal{N}(x; 1, 0.5^2) + 0.5\mathcal{N}(x; -1, 0.5^2),$$

where $\mathcal{N}(x; \mu, \sigma^2)$ is the density function of a $\mathcal{N}(\mu, \sigma^2)$ distribution. Using the standard substitution

$$\epsilon(x^k, k) = -\sigma_k \nabla \log p(x^k)$$

in the backward diffusion process, produces the backward diffusion process

$$x^{k-1} | x^k \sim \mathcal{N}((x^k + (1 - \alpha_k)\nabla \log p(x^k))/\sqrt{\alpha_k}, \sigma_k^2).$$

To illustrate the effects of a guidance function on the sampling process, we consider the guidance function associated with the (unscaled) score of a $\mathcal{N}(1, 0.5^2)$ distribution, i.e.

$$g(x) = -(x - 1)/0.5^2.$$

Then, the guided backward diffusion process has mean:

$$\frac{x^k + (1 - \alpha_k)\nabla \log p(x^k))}{\sqrt{\alpha_k}} + \sigma_k^2 g(x^k)$$

$$= \frac{x^k + (1 - \alpha_k)\left(\nabla \log p(x^k) + \sigma_k^2 g(x^k)\sqrt{\alpha_k}/(1 - \alpha_k)\right)}{\sqrt{\alpha_k}},$$

which corresponds to a standard backward diffusion process with the modified score function

$$\nabla \log p(x^k) + \sigma_k^2 g(x^k)\sqrt{\alpha_k}/(1 - \alpha_k),$$

which would place more weight on the rightmost mode of the Gaussian mixture during the backward diffusion process.

We run the backward denoising diffusion process using the exact score function $\nabla \log p(x^k)$. The sampling distributions of $x^k$ are plotted at various denoising time steps $k$ in Figure 3.

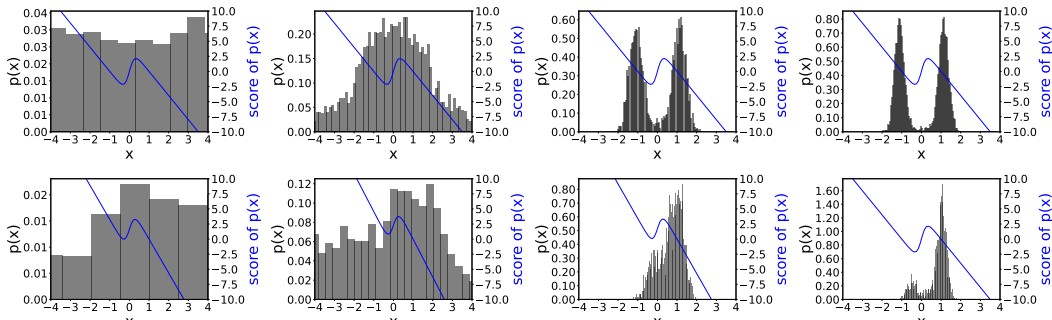

Figure 3: Pedagogical example illustrating guided diffusion sample generation for a Gaussian mixture $0.5\mathcal{N}(1, 0.5^2) + 0.5\mathcal{N}(-1, 0.5^2)$. **Top row:** histograms of samples from unguided backward diffusion at steps $k = 8, 6, 4, 0$, where $\nabla \log p(x)$ is the score of the Gaussian mixture shown in blue. **Bottom row:** histograms of samples from guided diffusion (2) using the score function of a $\mathcal{N}(1, 0.5^2)$ distribution, i.e. $g(x) = -(x - 1)/0.5^2$. The modified score function corresponding to the guided diffusion process is shown in blue. The guided score function (the score of the actual sampling density) is significantly shifted and skewed, relative to the original score function, at the intermediate denoising time steps ($k = 6, 4$). This ensures that the right mode of the Gaussian mixture is sampled more frequently during denoising.

# B  GaussianWorld Domain

The GaussianWorld domain is a toy 2-dimensional Markov decision process defined designed to illustrate and compare generalization and compositionality of diffusion models (Table 1). It is defined as follows:

- **Decision Epochs.** The decision epochs are $t = 0, 1, 2, \ldots T$ where we set $T = 128$ in our experiments.

- **State Space.** $\mathcal{S} = \mathbb{R}^2$ describes all positions $(x_t, y_t)$ of a particle in space at every decision epoch $t$. It is assumed that $x_t$ is the x-coordinate and $y_t$ is the y-coordinate. The initial state is $s_0 = (0, 0)$ unless otherwise specified.

- **Action Space.** $\mathcal{A} = \mathbb{R}$ describes the (counterclockwise) angle of the movement vector of the particle at every decision epoch, relative to the horizontal.

- **Transitions.** Letting $a_t$ be the angle of movement of the particle at time $t$, the transitions of $x_t$ and $y_t$ are defined as follows:

$$x_{t+1} = x_t + 0.02 \cdot \cos(a_t + \varepsilon_t), \qquad y_{t+1} = y_t + 0.02 \cdot \sin(a_t + \varepsilon_t), \qquad \varepsilon_t \sim \mathcal{N}(0, 0.2^2).$$

  Here, $\varepsilon_t$ is an i.i.d. Gaussian noise added to the actions before they are applied by the controller.

- **Reward Function and Discount.** The problem is not solved so we leave the reward unspecified. We also leave the discount factor unspecified.

## C   Proof that Conditional Diffusion Increases Entropy

We begin with the following definitions.

**Definition C.1** (Entropy). Let $p(x)$ be a density function of a random variable $X$ with support $\mathcal{X}$. The *entropy* of $X$ is defined as

$$H(X) = \int_{\mathcal{X}} p(x) \log \left( \frac{1}{p(x)} \right) \, \mathrm{d}x.$$

**Definition C.2** (Conditional Entropy). The *conditional entropy* of $X$ given $Y$ on support $\mathcal{Y}$ is defined as

$$H(X|Y) = \mathbb{E}_{y \in \mathcal{Y}}[H(X|Y = y)].$$

Our goal is to prove

**Theorem C.3.** *Let $S_t$ be the random state at time $t$ sampled according to the conditional distribution $p(S_{t+1} = s | S_t = x, A_t = u)$, and let $A_t$ be a random action following some conditional distribution $p(A_t = a | S_t = x)$. Then $H(\tau | S_t) \geq H(\tau | S_0, A_0 \ldots S_t)$, where $\tau$ is a (random) sub-trajectory beginning in state $S_t$.*

*Proof.* First, letting $U = (S_0, A_0, \ldots S_{t-1}, A_{t-1})$, observe that:

$$H(U, \tau | S_t = s)$$
$$= \iint p(U = u, \tau | S_t = s) \log \left( \frac{1}{p(U = u, \tau | S_t = s)} \right) \, \mathrm{d}u \, \mathrm{d}\tau$$
$$= \iint p(U = u, \tau | S_t = s) \log \left( \frac{1}{p(U = u | S_t = s) p(\tau | U = u, S_t = s)} \right) \, \mathrm{d}u \, \mathrm{d}\tau$$
$$= \int p(U = u | S_t = s) \log \left( \frac{1}{p(U = u | S_t = s)} \right) \, \mathrm{d}u$$
$$\quad + \int p(U = u | S_t = s) \int p(\tau | U = u, S_t = s) \log \left( \frac{1}{p(\tau | U = u, S_t = s)} \right) \, \mathrm{d}u \, \mathrm{d}\tau$$
$$= H(U | S_t = s) + \mathbb{E}_{u \in \mathcal{U} | S_t = s}[H(\tau | U = u, S_t = s)].$$

Next, using the additivity property of expectation and law of total expectation:

$$H(U, \tau | S_t) = \mathbb{E}_{s \in \mathcal{S}_t}[H(U | S_t = s)] + \mathbb{E}_{s \in \mathcal{S}_t, u \in \mathcal{U}}[H(\tau | U = u, S_t = s)]$$
$$= H(U | S_t) + H(\tau | U, S_t).$$

Next, we prove sub-additivity of conditional entropy:

$$H(U, \tau | S_t = s) - H(U | S_t = s) - H(\tau | S_t = s)$$

$$= \iint p(U = u, \tau | S_t = s) \log \left( \frac{1}{p(U = u, \tau | S_t = s)} \right) du \, d\tau$$

$$- \int p(U = u | S_t = s) \log \left( \frac{1}{p(U = u | S_t = s)} \right) du - \int p(\tau | S_t = s) \log \left( \frac{1}{p(\tau | S_t = s)} \right) d\tau$$

$$= \iint p(U = u, \tau | S_t = s) \log \left( \frac{p(\tau | S_t = s) p(U = u | S_t = s)}{p(U = u, \tau | S_t = s)} \right) du \, d\tau$$

$$\leq \log \iint p(U = u, \tau | S_t = s) \left( \frac{p(\tau | S_t = s) p(U = u | s_t = s)}{p(U = u, \tau | S_t = s)} \right) du \, d\tau$$

$$= \log 1 = 0,$$

where the inequality in the derivation follows by Jensen's inequality. This implies that

$$H(U, \tau | S_t = s) \leq H(U | S_t = s) + H(\tau | S_t = s).$$

Taking expectation of both sides with respect to $S_t$, and using the monotonicity and additivity properties of expectation:

$$H(U, \tau | S_t) = \mathbb{E}_{s \in \mathcal{S}_t}[H(U, \tau | S_t = s)]$$

$$\leq \mathbb{E}_{s \in \mathcal{S}_t}[H(U | S_t = s) + H(\tau | S_t = s)] = H(U | S_t) + H(\tau | S_t).$$

Finally, putting it all together:

$$H(\tau | U, S_t) = H(U, \tau | S_t) - H(U | S_t) \leq H(U | S_t) + H(\tau | S_t) - H(U | S_t) = H(\tau | S_t),$$

which completes the proof. □

## D    Theoretical Analysis

### D.1    Assumptions and Definitions

We decompose a full trajectory of length $T$ into $N = T/w$ non-overlapping sub-trajectories (or chunks), each of length $w$. Each *chunk* $S_i \in \mathcal{T}^{(w)}$ is defined as

$$S_i := (s_{iw}, a_{iw}, s_{iw+1}, a_{iw+1}, \ldots, s_{(i+1)w}).$$

Let the *full trajectory* be defined as

$$S = (S_0, S_1, \ldots, S_{N-1}).$$

We define the *boundary state* $X_i$ as the initial state of chunk $S_i$:

$$X_i := s_{iw}, \quad i = 0, 1 \ldots N,$$

which form the backbone of the generative process.

We assume the following factored generative process for trajectories

$$p(S_0, S_1, \ldots, S_{N-1}) = p(X_0) \prod_{i=0}^{N-1} p(S_i \mid X_i) \, p(X_{i+1} \mid S_i).$$

This implies that the boundary state sequence $X = (X_0, X_1, \ldots, X_N)$ forms a first-order Markov chain

$$p(X_{i+1} \mid S_i) = p(X_{i+1} \mid X_i).$$

Each chunk $S_i$ produces a scalar discounted return $Y_i$, defined as

$$Y_i := f(S_i) = \sum_{j=0}^{w-1} \gamma^j \hat{R}(s_{iw+j}, a_{iw+j}),$$

where $\hat{R}$ is a learned reward model, and $\gamma \in [0, 1]$ is the discount factor.

Given a bound $R_{\max} < \infty$ on the absolute reward, we define the *maximum per-chunk return bound* as:

$$B_w := \sum_{j=0}^{w-1} \gamma^j R_{\max} = \frac{R_{\max}(1 - \gamma^w)}{1 - \gamma} \quad \Rightarrow \quad |Y_i| \leq B_w.$$

The cumulative return over the full trajectory is approximated by

$$\hat{J} = \sum_{i=0}^{N-1} \gamma^{iw} Y_i,$$

and the expected return under the target policy $\pi$ is:

$$J(\pi) := \mathbb{E}_{p_\pi}[\hat{J}] = \mathbb{E}_{p_\pi}\left[ \sum_{i=0}^{N-1} \gamma^{iw} Y_i \right].$$

**Definition D.1** (Chunked Behavior Distributions). Let $p_\beta^{(w)}$ denote the true distribution over behavior chunks $S_i$, and let $\hat{p}_\beta^{(w)}$ be the learned conditional distribution modeled by the diffusion process. These distributions describe how chunks are generated given boundary states:

$$p_\beta^{(w)}(S_i \mid X_i), \qquad \hat{p}_\beta^{(w)}(S_i \mid X_i).$$

**Definition D.2** (Total Variation Distance). The *total variation distance* between two probability distributions $P$ and $Q$ over the same measurable space $\mathcal{X}$ is defined as

$$\mathrm{TV}(P, Q) := \sup_{A \subseteq \mathcal{X}} |P(A) - Q(A)|.$$

We now restate the two assumptions presented in the main text for convenience.

**Assumption D.3** (Bounded Likelihood Ratio). There is a constant $\kappa$ such that $\frac{\pi(a|s)}{\beta(a|s)} \leq \kappa$ for all $s \in \mathcal{S}$ and $a \in \mathcal{A}$.

Note that this assumption can be easily verified in our experimental setting. Since the action spaces are closed intervals and the behavior and target policy distributions are both represented as truncated Gaussian distributions, the ratio of the two policies is bounded over the action space.

**Assumption D.4** (Chunk-wise Model Fit). The total variation distance between the true chunk distribution $p_\beta^{(w)}$ and the learned conditional distribution $\hat{p}_\beta^{(w)}$ is bounded by some constant $\delta_\beta > 0$,

$$\mathrm{TV}\left(p_\beta^{(w)}, \hat{p}_\beta^{(w)}\right) \leq \delta_\beta.$$

### D.2 Analysis of the Bias

We begin by bounding the total variation distance between the true target distribution $p_\pi^{(w)}$ and the guided model $\hat{p}_\pi^{(w)}$.

**Lemma D.5.** *The total variation distance between the guided model $\hat{p}_\pi^{(w)}$ and the true target distribution $p_\pi^{(w)}$ satisfies*

$$\mathrm{TV}\left(p_\pi^{(w)}, \hat{p}_\pi^{(w)}\right) \leq \kappa^2 \cdot \delta_\beta$$

*Proof.* By the definition of total variation distance

$$\mathrm{TV}(p_\pi^{(w)}, \hat{p}_\pi^{(w)}) = \frac{1}{2} \int \left| p_\pi^{(w)}(\tau) - \hat{p}_\pi^{(w)}(\tau) \right| d\tau.$$

Using the reweighted form of each distribution

$$\mathrm{TV}(p_\pi^{(w)}, \hat{p}_\pi^{(w)}) = \frac{1}{2} \int \left| \left( p_\beta^{(w)}(\tau) - \hat{p}_\beta^{(w)}(\tau) \right) \cdot \prod_{j=0}^{w-1} \frac{\pi(a_j \mid s_j)}{\beta(a_j \mid s_j)} \right| d\tau,$$

and applying the bound on the likelihood ratio (Assumption D.3):

$$\mathrm{TV}(p_\pi^{(w)}, \hat{p}_\pi^{(w)}) \leq \frac{\kappa^w}{2} \int \left| p_\beta^{(w)}(\tau) - \hat{p}_\beta^{(w)}(\tau) \right| \mathrm{d}\tau = \kappa^w \cdot \mathrm{TV}(p_\beta^{(w)}, \hat{p}_\beta^{(w)}) \leq \kappa^w \cdot \delta_\beta.$$

This completes the proof. $\qquad\square$

Let the total variation distance between the true target distribution and the guided diffusion model be denoted by

$$\delta_\pi := \mathrm{TV}\left( p_\pi^{(w)}, \hat{p}_\pi^{(w)} \right).$$

By Lemma D.5, we have the bound

$$\delta_\pi \leq \kappa^w \cdot \delta_\beta.$$

We now derive a bound on the absolute bias of the estimated return when sampling chunks from the guided model $\hat{p}_\pi^{(w)}$ instead of the true target distribution $p_\pi^{(w)}$.

**Lemma D.6** (Expectation Difference Bound via Total Variation). *Let $p$ and $q$ be two probability densities on a probability space $\mathcal{X}$. Let*

$$\|f\|_\infty = \sup_{x \in \mathcal{X}} |f(x)|$$

*be the supremum norm of a bounded function $f : \mathcal{X} \to \mathbb{R}$, and let:*

$$\|p - q\|_1 = \int_{\mathcal{X}} |p(x) - q(x)| \, \mathrm{d}x, \qquad \mathrm{TV}(p, q) = \tfrac{1}{2} \|p - q\|_1.$$

*Then*

$$\left| \mathbb{E}_{x \sim p}[f(x)] - \mathbb{E}_{x \sim q}[f(x)] \right| \leq 2 \|f\|_\infty \, \mathrm{TV}(p, q).$$

*Proof.*

$$\left| \mathbb{E}_p[f] - \mathbb{E}_q[f] \right| = \left| \int_{\mathcal{X}} f(x) \, p(x) \, \mathrm{d}x - \int_{\mathcal{X}} f(x) \, q(x) \, \mathrm{d}x \right|$$

$$= \left| \int_{\mathcal{X}} f(x) \left( p(x) - q(x) \right) \mathrm{d}x \right| \leq \int_{\mathcal{X}} |f(x)| \, |p(x) - q(x)| \, \mathrm{d}x$$

$$\leq \|f\|_\infty \int_{\mathcal{X}} |p(x) - q(x)| \, \mathrm{d}x = 2 \|f\|_\infty \, \mathrm{TV}(p, q).$$

This completes the proof. $\qquad\square$

**Lemma D.7** (Marginal TV Bound via Conditional TV). *Let $p(x \mid s)$ and $\hat{p}(x \mid s)$ be conditional densities over chunk $x \in \mathcal{T}^{(w)}$, given state $s \in \mathcal{S}$, and let $\mu(s)$ denote the marginal distribution over $s$. Then*

$$\mathrm{TV}\left( \int p(x \mid s)\mu(s)\mathrm{d}s, \int \hat{p}(x \mid s)\mu(s)\mathrm{d}s \right) \leq \int \mathrm{TV}\left( p(\cdot \mid s), \hat{p}(\cdot \mid s) \right) \mu(s)\mathrm{d}s.$$

*In particular, if $\mathrm{TV}(p(\cdot \mid s), \hat{p}(\cdot \mid s)) \leq \epsilon$ for all $s$, then*

$$\mathrm{TV}(p, \hat{p}) \leq \epsilon.$$

*Proof.* Let $p(x) = \int p(x \mid s)\mu(s)\mathrm{d}s$, $\hat{p}(x) = \int \hat{p}(x \mid s)\mu(s)\mathrm{d}s$. Then:

$$\mathrm{TV}(p, \hat{p}) = \frac{1}{2} \int |p(x) - \hat{p}(x)| \, \mathrm{d}x$$

$$= \frac{1}{2} \int \left| \int \mu(s) \left[ p(x \mid s) - \hat{p}(x \mid s) \right] \mathrm{d}s \right| \mathrm{d}x$$

$$\leq \frac{1}{2} \iint \mu(s) |p(x \mid s) - \hat{p}(x \mid s)| \, \mathrm{d}s \, \mathrm{d}x \quad \text{(by Jensen's inequality)}$$

$$= \int \mu(s) \left[ \frac{1}{2} \int |p(x \mid s) - \hat{p}(x \mid s)| \, \mathrm{d}x \right] \mathrm{d}s$$

$$= \int \mu(s) \cdot \mathrm{TV}(p(\cdot \mid s), \hat{p}(\cdot \mid s))\mathrm{d}s.$$

If $\mathrm{TV}(p(\cdot \mid s), \hat{p}(\cdot \mid s)) \leq \epsilon$ uniformly, the integral is bounded by $\epsilon$. $\qquad\square$

**Theorem D.8** (Bias Bound for STITCH-OPE). *The bias of the return estimate under the guided diffusion model satisfies*

$$\left| \mathbb{E}_{\hat{p}_\pi}[\hat{J}] - J(\pi) \right| \leq \frac{2B_w}{1 - \gamma^w} \cdot \delta_\pi.$$

*Proof.* The return estimator is:

$$\hat{J} = \sum_{i=0}^{N-1} \gamma^{iw} Y_i, \qquad \text{where} \quad Y_i = f(S_i) = \sum_{j=0}^{w-1} \gamma^j \hat{R}(s_{iw+j}, a_{iw+j}).$$

Thus, the bias is:

$$\left| \mathbb{E}_{\hat{p}_\pi}[\hat{J}] - \mathbb{E}_{p_\pi}[\hat{J}] \right| = \left| \sum_{i=0}^{N-1} \gamma^{iw} \left( \mathbb{E}_{\hat{p}_\pi}[Y_i] - \mathbb{E}_{p_\pi}[Y_i] \right) \right| \leq \sum_{i=0}^{N-1} \gamma^{iw} \left| \mathbb{E}_{\hat{p}_\pi}[Y_i] - \mathbb{E}_{p_\pi}[Y_i] \right|.$$

For each chunk $i$, $Y_i$ depends only on $S_i$, with marginal distributions $\hat{p}_\pi^{(w,i)}$ and $p_\pi^{(w,i)}$ under $\hat{p}_\pi$ and $p_\pi$, respectively. By Lemma D.6 and Lemma D.7

$$|\mathbb{E}_{\hat{p}_\pi}[Y_i] - \mathbb{E}_{p_\pi}[Y_i]| \leq 2 \cdot \sup |Y_i| \cdot \text{TV}(p_\pi^{(w,i)}, \hat{p}_\pi^{(w,i)}).$$

Since $|\hat{R}(s,a)| \leq R_{\max}$, the per-chunk return is bounded:

$$|Y_i| \leq \sum_{j=0}^{w-1} \gamma^j R_{\max} = R_{\max} \cdot \frac{1 - \gamma^w}{1 - \gamma}.$$

Using Lemma D.5, we know that $\text{TV}(p_\pi^{(w,i)}, \hat{p}_\pi^{(w,i)}) \leq \delta_\pi$, Thus we have

$$|\mathbb{E}_{\hat{p}_\pi}[Y_i] - \mathbb{E}_{p_\pi}[Y_i]| \leq 2 \cdot \frac{R_{\max}(1 - \gamma^w)}{1 - \gamma} \cdot \delta_\pi.$$

Summing over chunks:

$$\left| \mathbb{E}_{\hat{p}_\pi}[\hat{J}] - \mathbb{E}_{p_\pi}[\hat{J}] \right| \leq \sum_{i=0}^{N-1} \gamma^{iw} \cdot 2 \cdot \frac{R_{\max}(1 - \gamma^w)}{1 - \gamma} \cdot \delta_\pi = 2 \cdot \frac{R_{\max}(1 - \gamma^w)}{1 - \gamma} \cdot \delta_\pi \cdot \sum_{i=0}^{N-1} \gamma^{iw}.$$

The geometric sum is:

$$\sum_{i=0}^{N-1} \gamma^{iw} \leq \sum_{i=0}^{\infty} \gamma^{iw} = \frac{1}{1 - \gamma^w}.$$

Thus:

$$\left| \mathbb{E}_{\hat{p}_\pi}[\hat{J}] - \mathbb{E}_{p_\pi}[\hat{J}] \right| \leq 2 \cdot \frac{R_{\max}(1 - \gamma^w)}{1 - \gamma} \cdot \delta_\pi \cdot \frac{1}{1 - \gamma^w} = \frac{2B_w}{1 - \gamma^w} \cdot \delta_\pi.$$

This completes the proof. $\qquad\qquad\qquad\qquad\qquad\qquad\qquad\qquad\qquad\qquad\qquad\qquad\square$

**Corollary D.9** (Bias Bound in Terms of Model Fit $\delta_\beta$). *Under the assumptions* $\sup_i \text{TV}(p_\pi^{(w,i)}, \hat{p}_\pi^{(w,i)}) \leq \delta_\pi \leq \kappa^w \cdot \delta_\beta$ *and* $\sup_\tau |\hat{J}(\tau)| \leq \frac{R_{\max}}{1-\gamma}$, *the bias satisfies*

$$\left| \mathbb{E}_{\hat{p}_\pi}[\hat{J}] - J(\pi) \right| \leq \frac{2B_w}{1 - \gamma^w} \cdot \kappa^w \cdot \delta_\beta.$$

### D.3 Analysis of the Variance

**Lemma D.10** (Conditional Independence of Chunk Rewards). *Let* $X_i := s_{iw}$ *be the boundary state at the start of chunk* $S_i$, *and define:*

$$Y_i := f(S_i) = \sum_{j=0}^{w-1} \gamma^j \, \hat{R}(s_{iw+j}, a_{iw+j}).$$

*Assume the generative process satisfies the following properties:*

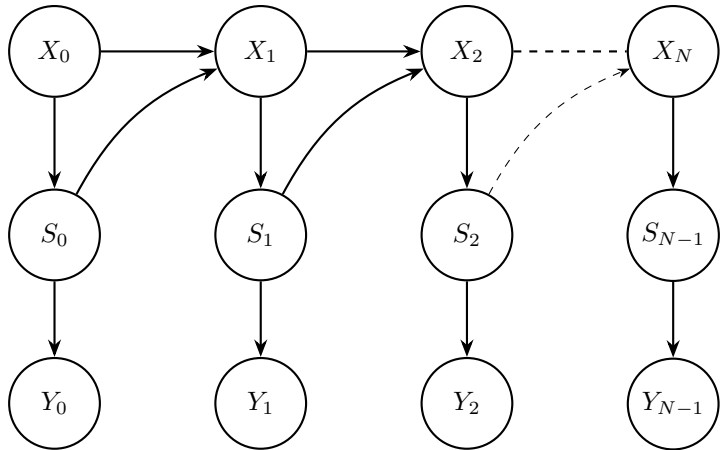

Figure 4: Illustration of the sub-trajectory decomposition. Each chunk $S_i$ generates a reward sequence $Y_i$ and leads to a boundary state $X_{i+1}$.

- *Each chunk $S_i$ is generated independently given $X_i$*

- *The return $Y_i$ is a deterministic function of $S_i$.*

*Then for all $i \neq j$, the returns $Y_i$ and $Y_j$ are conditionally independent given the full boundary state chain $X_0, X_1, \ldots, X_N$,*

$$Y_i \perp\!\!\!\perp Y_j \mid X_0, \ldots, X_N.$$

*Proof.* Refer to the graphical model in Figure 4. The nodes $X_0, X_1, \ldots, X_N$ form a Markov chain. Each chunk $S_i$ is a child of $X_i$, and each return $Y_i$ is a child of $S_i$.

Now consider any path from $Y_i$ to $Y_j$. Such a path must go through:

$$Y_i \leftarrow S_i \leftarrow X_i \rightsquigarrow X_{i+1} \rightsquigarrow \cdots \rightsquigarrow X_j \rightarrow S_j \rightarrow Y_j.$$

All such paths must traverse through at least one boundary node $X_k$. Since we are conditioning on all $X_0, \ldots, X_N$, and these nodes are non-colliders on every path from $Y_i$ to $Y_j$, all such paths are blocked. By the criterion of d-separation (see, e.g. Chapter 8 in [49]), this implies $Y_i \perp\!\!\!\perp Y_j \mid X_0, \ldots, X_N$. $\square$

**Theorem D.11** (Variance Bound). *Let $\hat{p}_\pi$ denote the trajectory distribution induced by the guided diffusion model, and $p_\pi$ the true trajectory distribution under the target policy. Let $\hat{J}$ be the return estimator using a learned reward model. Then*

$$\mathrm{Var}_{\hat{p}_\pi}(\hat{J}) \leq \mathrm{Var}_{p_\pi}(J) + 10 \left(\frac{T}{w}\right)^2 B_w^2 \kappa^w \delta_\beta + \frac{2 B_w^2}{1 - \gamma^{2w}} \kappa^w \delta_\beta,$$

*where $B_w$ denotes the maximum per-chunk discounted return.*

*Proof.* We begin by applying the law of total variance under the guided model distribution $\hat{p}_\pi$

$$\mathrm{Var}_{\hat{p}_\pi}(\hat{J}) = \mathbb{E}_{\hat{p}_\pi}\left[\mathrm{Var}_{\hat{p}_\pi}(\hat{J} \mid X)\right] + \mathrm{Var}_{\hat{p}_\pi}\left(\mathbb{E}_{\hat{p}_\pi}[\hat{J} \mid X]\right).$$

Using Lemma D.10 we have that the chunk-level rewards $Y_i$ and $Y_j$ are conditionally independent given the boundary states $X_0, X_1, \ldots, X_N$:

$$Y_i \perp\!\!\!\perp Y_j \mid X_0, X_1, \ldots, X_N \qquad \text{for all } i \neq j.$$

Using this conditional independence, the variance of the total return under $\hat{p}_\pi$ factorizes:

$$\mathrm{Var}_{\hat{p}_\pi}[\hat{J} \mid X] = \mathrm{Var}_{\hat{p}_\pi}\left[\sum_{i=0}^{N-1} \gamma^{iw} Y_i \,\middle|\, X\right] = \sum_{i=0}^{N-1} \gamma^{2iw} \cdot \mathrm{Var}_{\hat{p}_\pi}(Y_i \mid X_i).$$

To bound the difference in conditional variances, we apply the law of variance

$$\text{Var}(Y_i \mid X_i) = \mathbb{E}[Y_i^2 \mid X_i] - (\mathbb{E}[Y_i \mid X_i])^2.$$

Let us define a bound on the per-chunk return magnitude:

$$B_w := \frac{R_{\max}(1 - \gamma^w)}{1 - \gamma} \quad \Rightarrow \quad |Y_i| \leq B_w, \quad Y_i^2 \leq B_w^2.$$

Using Lemma D.6 (Expectation Difference Bound via Total Variation), we have

$$|\mathbb{E}_{p_\pi}[f] - \mathbb{E}_{\hat{p}_\pi}[f]| \leq 2\delta_\pi \cdot \|f\|_\infty.$$

Applying this with $f = Y_i$ and $f = Y_i^2$, and using the bound $|Y_i| \leq B_w$, we obtain:

$$|\mathbb{E}_{p_\pi}[Y_i] - \mathbb{E}_{\hat{p}_\pi}[Y_i]| \leq 2\delta_\pi B_w, \qquad \left|\mathbb{E}_{p_\pi}[Y_i^2] - \mathbb{E}_{\hat{p}_\pi}[Y_i^2]\right| \leq 2\delta_\pi B_w^2.$$

We analyze the difference in conditional variances:

$$\begin{aligned}
&\left|\text{Var}_{\hat{p}_\pi}(Y_i \mid X_i) - \text{Var}_{p_\pi}(Y_i \mid X_i)\right| \\
&= \left|\mathbb{E}_{\hat{p}_\pi}[Y_i^2] - \mathbb{E}_{p_\pi}[Y_i^2] - \left(\mathbb{E}_{\hat{p}_\pi}[Y_i]^2 - \mathbb{E}_{p_\pi}[Y_i]^2\right)\right| \\
&\leq \left|\mathbb{E}_{\hat{p}_\pi}[Y_i^2] - \mathbb{E}_{p_\pi}[Y_i^2]\right| + \left|\mathbb{E}_{\hat{p}_\pi}[Y_i]^2 - \mathbb{E}_{p_\pi}[Y_i]^2\right| \\
&= \left|\mathbb{E}_{\hat{p}_\pi}[Y_i^2] - \mathbb{E}_{p_\pi}[Y_i^2]\right| + |\mathbb{E}_{\hat{p}_\pi}[Y_i] - \mathbb{E}_{p_\pi}[Y_i]| \cdot |\mathbb{E}_{\hat{p}_\pi}[Y_i] + \mathbb{E}_{p_\pi}[Y_i]| \\
&\leq 2\delta_\pi B_w^2 + (2\delta_\pi B_w)(2B_w) \\
&= 6\delta_\pi B_w^2.
\end{aligned}$$

This uses the triangle inequality and the identity $|a^2 - b^2| = |a - b||a + b|$, along with the bounds $|Y_i| \leq B_w$, $\|Y_i\|_\infty^2 \leq B_w^2$, and total variation guarantees from Lemma D.6. Then

$$\left|\text{Var}_{\hat{p}_\pi}(Y_i \mid X_i) - \text{Var}_{p_\pi}(Y_i \mid X_i)\right| \leq 6\delta_\pi B_w^2.$$

We now return to bounding the first term in the law of total variance

$$\mathbb{E}_{\hat{p}_\pi}\left[\text{Var}_{\hat{p}_\pi}(\hat{J} \mid X)\right] = \mathbb{E}_{\hat{p}_\pi}\left[\sum_{i=0}^{N-1} \gamma^{2iw} \cdot \text{Var}_{\hat{p}_\pi}(Y_i \mid X_i)\right].$$

Using the bound from the previous step

$$\text{Var}_{\hat{p}_\pi}(Y_i \mid X_i) \leq \text{Var}_{p_\pi}(Y_i \mid X_i) + 6\delta_\pi B_w^2.$$

Taking expectation over $\hat{p}_\pi$ on both sides

$$\mathbb{E}_{\hat{p}_\pi}\left[\text{Var}_{\hat{p}_\pi}(Y_i \mid X_i)\right] \leq \mathbb{E}_{\hat{p}_\pi}\left[\text{Var}_{p_\pi}(Y_i \mid X_i)\right] + 6\delta_\pi B_w^2.$$

Now, using the expectation difference bound from Lemma D.6 again:

$$|\mathbb{E}_{\hat{p}_\pi}[f] - \mathbb{E}_{p_\pi}[f]| \leq 2\delta_\pi\|f\|_\infty, \qquad \text{where} \quad f(X_i) := \text{Var}_{p_\pi}(Y_i \mid X_i) \leq B_w^2.$$

So

$$\mathbb{E}_{\hat{p}_\pi}\left[\text{Var}_{p_\pi}(Y_i \mid X_i)\right] \leq \mathbb{E}_{p_\pi}\left[\text{Var}_{p_\pi}(Y_i \mid X_i)\right] + 2\delta_\pi B_w^2.$$

Combining both components

$$\mathbb{E}_{\hat{p}_\pi}\left[\text{Var}_{\hat{p}_\pi}(Y_i \mid X_i)\right] \leq \mathbb{E}_{p_\pi}\left[\text{Var}_{p_\pi}(Y_i \mid X_i)\right] + 8\delta_\pi B_w^2.$$

Summing across all chunks:

$$\begin{aligned}
\mathbb{E}_{\hat{p}_\pi}\left[\text{Var}_{\hat{p}_\pi}(\hat{J} \mid X)\right] &= \sum_{i=0}^{N-1} \gamma^{2iw} \cdot \mathbb{E}_{\hat{p}_\pi}\left[\text{Var}_{\hat{p}_\pi}(Y_i \mid X_i)\right] \\
&\leq \sum_{i=0}^{N-1} \gamma^{2iw}\left(\mathbb{E}_{p_\pi}\left[\text{Var}_{p_\pi}(Y_i \mid X_i)\right] + 8\delta_\pi B_w^2\right).
\end{aligned}$$

We can split the sum and factor out constants:

$$\mathbb{E}_{\hat{p}_\pi}\left[\mathrm{Var}_{\hat{p}_\pi}(\hat{J}\mid X)\right] = \sum_{i=0}^{N-1}\gamma^{2iw}\cdot\mathbb{E}_{p_\pi}\left[\mathrm{Var}_{p_\pi}(Y_i\mid X_i)\right] + 8\delta_\pi B_w^2\sum_{i=0}^{N-1}\gamma^{2iw}.$$

Let us define the chunk-level return variance

$$\mathbb{E}_{p_\pi}\left[\mathrm{Var}_{p_\pi}(\hat{J}\mid X)\right] := \sum_{i=0}^{N-1}\gamma^{2iw}\cdot\mathbb{E}_{p_\pi}\left[\mathrm{Var}_{p_\pi}(Y_i\mid X_i)\right].$$

Therefore

$$\boxed{\mathbb{E}_{\hat{p}_\pi}\left[\mathrm{Var}_{\hat{p}_\pi}(\hat{J}\mid X)\right] \le \mathbb{E}_{p_\pi}\left[\mathrm{Var}_{p_\pi}(\hat{J}\mid X)\right] + \frac{8\delta_\pi B_w^2}{1-\gamma^{2w}}.}$$

To complete the law of total variance, we now analyze the second term:

$$\mathrm{Var}_{\hat{p}_\pi}\left(\mathbb{E}_{\hat{p}_\pi}[\hat{J}\mid X]\right) = \mathrm{Var}_{\hat{p}_\pi}(Z_{\hat{p}}), \quad \text{where } Z_{\hat{p}} := \sum_{k=0}^{N-1}g_k(X_k), \quad g_k(x) := \mathbb{E}_{\hat{p}_\pi}[Y_k\mid X_k = x].$$

We define the corresponding ideal (true model) version:

$$Z_p := \sum_{k=0}^{N-1}\tilde{g}_k(X_k), \qquad \tilde{g}_k(x) := \mathbb{E}_{p_\pi}[Y_k\mid X_k = x].$$

Our goal is to bound the variance difference:

$$\Delta_{\mathrm{mean}} := \mathrm{Var}_{\hat{p}_\pi}(Z_{\hat{p}}) - \mathrm{Var}_{p_\pi}(Z_p) = (M_{\hat{p}} - M_p) - (m_{\hat{p}} - m_p)(m_{\hat{p}} + m_p),$$

where $M_{\hat{p}} := \mathbb{E}_{\hat{p}_\pi}[Z_{\hat{p}}^2]$, $m_{\hat{p}} := \mathbb{E}_{\hat{p}_\pi}[Z_{\hat{p}}]$, and similarly for $M_p$, $m_p$.

Insert and subtract a common term:

$$m_{\hat{p}} - m_p = \sum_{k=0}^{N-1}\left(\mathbb{E}_{\hat{p}_\pi}[g_k(X_k)] - \mathbb{E}_{\hat{p}_\pi}[\tilde{g}_k(X_k)]\right) + \sum_{k=0}^{N-1}\left(\mathbb{E}_{\hat{p}_\pi}[\tilde{g}_k(X_k)] - \mathbb{E}_{p_\pi}[\tilde{g}_k(X_k)]\right).$$

Each term is bounded by $2\delta_\pi B_w$, so $|m_{\hat{p}} - m_p| \le 4N\delta_\pi B_w$.

Expand both squares:

$$Z_{\hat{p}}^2 = \sum_{k=0}^{N-1}g_k^2(X_k) + 2\sum_{0\le k<\ell\le N-1}g_k(X_k)g_\ell(X_\ell),$$

$$Z_p^2 = \sum_{k=0}^{N-1}\tilde{g}_k^2(X_k) + 2\sum_{0\le k<\ell\le N-1}\tilde{g}_k(X_k)\tilde{g}_\ell(X_\ell).$$

Each term (both diagonal and cross terms) is bounded in total variation with sup-norm $B_w^2$, yielding

$$|M_{\hat{p}} - M_p| \le 2N^2\delta_\pi B_w^2.$$

From the bound on the means:

$$|m_{\hat{p}}|, |m_p| \le NB_w \quad\Rightarrow\quad |m_{\hat{p}} + m_p| \le 2NB_w.$$

So, the product term:

$$|(m_{\hat{p}} - m_p)(m_{\hat{p}} + m_p)| \le (4N\delta_\pi B_w)(2NB_w) = 8N^2\delta_\pi B_w^2.$$

Combining both:

$$|\Delta_{\mathrm{mean}}| = |\mathrm{Var}_{\hat{p}_\pi}(Z_{\hat{p}}) - \mathrm{Var}_{p_\pi}(Z_p)| \le 2N^2\delta_\pi B_w^2 + 8N^2\delta_\pi B_w^2 = 10N^2\delta_\pi B_w^2,$$

which yields

$$\boxed{\left|\mathrm{Var}_{\hat{p}_\pi}\left(\mathbb{E}_{\hat{p}_\pi}[\hat{J}\mid X]\right) - \mathrm{Var}_{p_\pi}\left(\mathbb{E}_{p_\pi}[J\mid X]\right)\right| \le 10\cdot\frac{T^2}{w^2}\cdot\delta_\pi B_w^2.}$$

Combining the two components from the law of total variance, we conclude:

$$\text{Var}_{\hat{p}_\pi}(\hat{J}) = \mathbb{E}_{\hat{p}_\pi}\left[\text{Var}_{\hat{p}_\pi}(\hat{J} \mid X)\right] + \text{Var}_{\hat{p}_\pi}\left(\mathbb{E}_{\hat{p}_\pi}[\hat{J} \mid X]\right)$$

$$\leq \mathbb{E}_{p_\pi}\left[\text{Var}_{p_\pi}(\hat{J} \mid X)\right] + \frac{8\delta_\pi B_w^2}{1 - \gamma^{2w}} + \text{Var}_{p_\pi}\left(\mathbb{E}_{p_\pi}[J \mid X]\right) + 10\left(\frac{T}{w}\right)^2 \delta_\pi B_w^2$$

$$= \text{Var}_{p_\pi}(J) + 10\left(\frac{T}{w}\right)^2 \delta_\pi B_w^2 + \frac{8\delta_\pi B_w^2}{1 - \gamma^{2w}}.$$

By Lemma D.5,

$$\boxed{\text{Var}_{\hat{p}_\pi}(\hat{J}) \leq \text{Var}_{p_\pi}(J) + 10\left(\frac{T}{w}\right)^2 B_w^2 \kappa^w \delta_\beta + \frac{8B_w^2}{1 - \gamma^{2w}}\kappa^w \delta_\beta,}$$

and the proof is complete. □

### D.4 Proof of the Bias-Variance Decomposition (Theorem 3.3)

Finally, we can bound the mean squared error of STITCH-OPE.

**Theorem D.12.** *Under Assumption D.3 and D.4, and using the notation of Theorem D.8 and Theorem D.11, the mean squared error of STITCH-OPE is bounded by*

$$\mathbb{E}_{\hat{p}_\pi}\left[(\hat{J} - J(\pi))^2\right] \leq \left(\frac{2B_w}{1 - \gamma^w}\kappa^w \delta_\beta\right)^2 + 10\left(\frac{T}{w}\right)^2 B_w^2 \kappa^w \delta_\beta + \frac{8B_w^2}{1 - \gamma^{2w}}\kappa^w \delta_\beta + \text{Var}_{p_\pi}(J).$$

*Proof.* We start by adapting the standard bias-variance decomposition to our setting:

$$\mathbb{E}_{\hat{p}_\pi}\left[(\hat{J} - J(\pi))^2\right] = \mathbb{E}_{\hat{p}_\pi}\left[(\hat{J} - \mathbb{E}_{\hat{p}_\pi}[\hat{J}] + \mathbb{E}_{\hat{p}_\pi}[\hat{J}] - J(\pi))^2\right]$$

$$= \mathbb{E}_{\hat{p}_\pi}\left[(\hat{J} - \mathbb{E}_{\hat{p}_\pi}[\hat{J}])^2\right] + \mathbb{E}_{\hat{p}_\pi}\left[(\mathbb{E}_{\hat{p}_\pi}[\hat{J}] - J(\pi))^2\right]$$

$$+ \mathbb{E}_{\hat{p}_\pi}\left[(\hat{J} - \mathbb{E}_{\hat{p}_\pi}[\hat{J}])(\mathbb{E}_{\hat{p}_\pi}[\hat{J}] - J(\pi))\right]$$

$$= \text{Var}_{\hat{p}_\pi}(\hat{J}) + \text{Bias}_{\hat{p}_\pi}(\hat{J})^2 + (\mathbb{E}_{\hat{p}_\pi}[\hat{J}] - J(\pi))(\mathbb{E}_{\hat{p}_\pi}[\hat{J} - \mathbb{E}_{\hat{p}_\pi}[\hat{J}]])$$

$$= \text{Var}_{\hat{p}_\pi}(\hat{J}) + \text{Bias}_{\hat{p}_\pi}(\hat{J})^2,$$

since the last term is zero. Plugging in the bounds of Theorems D.8 and D.11 completes the proof. □

## E  Pseudocode

A high-level pseudocode of conditional diffusion model training in STITCH-OPE is provided as Algorithm 1. A pseudocode of the off-policy evaluation subroutine for a single rollout is provided as Algorithm 2. Empirically, we have found that per-term normalization of the guidance function (line 9) resulted in more consistent performance, and allowed the guidance coefficients $\alpha$ and $\lambda$ to be more easily tuned.

## F  Domains

We include experiments on the medium datasets from the D4RL offline suite [13], and Pendulum and Acrobot domains from the OpenAI Gym suite [4]. We set the evaluation horizon to $T = 768$ for D4RL, $T = 256$ for Acrobot and $T = 196$ for Pendulum, and we use $\gamma = 0.99$ in all experiments. Furthermore, Acrobot uses a discrete action space and is incompatible with our method, so we modified the domain to take continuous actions. Table 5 summarizes the key properties of each domain.

---

**Algorithm 1** Conditional Diffusion Model Training in STITCH-OPE

---

**Require:** diffusion model $\epsilon_\theta(\tau, k|s)$, behavior data $\mathcal{D}_\beta$, $w \geq 0$, learning rate $\eta > 0$, $\{\sigma_k\}_{k=1}^K$ and $\{\alpha_k\}_{k=1}^K$ positive

1: $\bar{\alpha}_k \leftarrow \prod_{t=1}^k \alpha_t$ **for** $k = 1 \ldots K$
2: **initialize** $\theta$ randomly
3: **repeat**
4:     **sample** length-$w$ sub-trajectory $\tau^0 = (s_0, a_0, s_1, \ldots s_w)$ from $\mathcal{D}_\beta$
5:     **sample** $k \sim \text{Uniform}(\{1, \ldots K\})$         ▷ Sample denoising time step $k$
6:     **sample** $\epsilon \sim \mathcal{N}(0, I)$         ▷ Sample pure noise sub-trajectory
7:     $\nabla_\theta \mathcal{L}(\theta) \leftarrow \nabla_\theta \|\epsilon - \epsilon_\theta(\sqrt{\bar{\alpha}_k}\tau^0 + \sigma_k \epsilon, k|s_0)\|^2$         ▷ Gradient descent step on $\theta$
8:     $\theta \leftarrow \theta - \eta \nabla_\theta \mathcal{L}(\theta)$
9: **until** converged
10: **return** $\epsilon_\theta$

---

---

**Algorithm 2** Off-Policy Evaluation in STITCH-OPE

---

**Require:** diffusion model $\epsilon_\theta(\tau, k|s)$ (Algorithm 1), empirical reward function $\hat{R}(s, a)$, behavior policy $\beta(a|s)$, target policy $\pi(a|s)$, $\alpha \geq 0$, $\lambda \geq 0$, $w \geq 0$ (divides $T$), $\{\sigma_k\}_{k=1}^K$ and $\{\alpha_k\}_{k=1}^K$ positive

1: $\hat{J} \leftarrow 0$
2: **sample** $s_0^0 \sim d_0$         ▷ Sample initial state
3: **for** $t = 0$ **to** $T/w - 1$ **do**         ▷ Generation for decision epochs $wt$ to $w(t+1)$
4:     **sample** $\tau_{wt:w(t+1)}^K \sim \mathcal{N}(0, I)$         ▷ Sample pure noise sub-trajectory
5:     **for** $k = K$ **to** $1$ **do**         ▷ Denoising step $k$
6:         $\mu_{k-1} \leftarrow \frac{1}{\sqrt{\alpha_k}}\left(\tau_{wt:w(t+1)}^k - \frac{1-\alpha_k}{\sigma_k}\epsilon_\theta(\tau_{wt:w(t+1)}^k, k \,|\, s_{wt}^0)\right)$         ▷ Mean of diffusion
7:         $g_k^\pi \leftarrow \sum_{u=wt}^{w(t+1)-1} \nabla_\tau \log \pi(a_u^k|s_u^k)$         ▷ Compute $\pi$ guidance term
8:         $g_k^\beta \leftarrow \sum_{u=wt}^{w(t+1)-1} \nabla_\tau \log \beta(a_u^k|s_u^k)$         ▷ Compute $\beta$ guidance term
9:         $g_k \leftarrow \alpha(g_k^\pi/\|g_k^\pi\|_2) - \lambda(g_k^\beta/\|g_k^\beta\|_2)$         ▷ Compute normalized guidance
10:         **sample** $\tau_{wt:w(t+1)}^{k-1} \sim \mathcal{N}\left(\mu_k + \sigma_k^2 g_k, \sigma_k^2 I\right)$         ▷ Apply guided diffusion step
11:     **end for**
12:     $\hat{J} \leftarrow \hat{J} + \sum_{u=wt}^{w(t+1)-1} \gamma^u \hat{R}(s_u^0, a_u^0)$         ▷ Update $\pi$ return using denoised $\tau_{wt:w(t+1)}^0$
13: **end for**
14: **return** $\hat{J}$

---

| Description | Hopper | Walker | HalfCheetah | Pendulum | Acrobot |
|---|---|---|---|---|---|
| state dimension | 11 | 17 | 17 | 3 | 6 |
| action dimension | 3 | 6 | 6 | 1 | 3 |
| range of action | $[-1, 1]$ | $[-1, 1]$ | $[-1, 1]$ | $[-2, 2]$ | $[-1, 1]$ |
| rollout length $T$ | 768 | 768 | 768 | 196 | 256 |
| discount factor $\gamma$ | 0.99 | 0.99 | 0.99 | 0.99 | 0.99 |

Table 5: Properties of D4RL [13] and OpenAI Gym [4] benchmark problems.

# G   Policies

**D4RL Offline Suite.** Behavior and target policies and their trained procedures are described in [14], and the policy parameters are borrowed from the official repository at `https://github.com/google-research/deep_ope` (Apache 2.0 licensed). The 10 target policies of varying ability, $\pi_{\theta_1}, \pi_{\theta_2}, \ldots \pi_{\theta_{10}}$, are obtained by checkpointing the policy parameters $\theta_1, \theta_2 \ldots \theta_{10}$ at various points during training. Each target policy network models the action probability distribution $\pi_i(a|s)$ using a set of independent Gaussian distributions, predicting the mean and variance $(\mu_i, \sigma_i^2)$ of each action component $a_i$ independently. This allows the score function of the target policy to be easily computed. As discussed in the main text, all policies are derived from the medium datasets in all experiments.

**OpenAI Gym.** We model target policies $\pi_1, \pi_2 \ldots \pi_5$ as MLPs and train them in each environment following the Twin-Delayed DDPG (TD3) [8] algorithm. The total training time is set to 50000 steps, and we checkpoint policies every 5000 steps. The behavior policy is set to the target policy $\pi_3$. The complete list of hyper-parameters is provided in Table 6.

| Description | Value |
|---|---|
| number of hidden layers in actor and critic | 2 |
| number of neurons per layer in actor and critic | 256 |
| hidden activation function | ReLU |
| output activation function | tanh |
| Gaussian noise for exploration | 0.1 |
| noise added to target policy during critic update | 0.2 |
| target noise clipping | 0.5 |
| frequency of delayed policy updates | 2 |
| moving average of target $\theta'$ | 0.005 |
| learning rate of Adam optimizer | 0.0003 |
| batch size | 256 |
| replay buffer size | 1000000 |

Table 6: Hyper-parameters for training target policies on OpenAI Gym domains.

**Bounded Action Space.** Since the action spaces for all domains are compact bounded intervals, we need to restrict the action space of the policy networks during evaluation. We accomplish this by applying the tanh transformation to each Gaussian action distribution and then scaling the result to the required range. Note that this transformation constrains the action probability distribution of all policies to a bounded range, and thus satisfies the requirement of Assumption 3.1.

## H   Baselines

The following model-free baseline methods were chosen for empirical comparison with STITCH-OPE:

**Fitted Q-Evaluation (FQE).** [22] evaluates a target policy $\pi$ by estimating its Q-value function $Q_\theta(s, a)$ using a neural network. The loss function for $\theta$ is

$$\mathcal{L}_{FQE}(\theta) = \mathbb{E}_{\substack{(s,a,r,s') \sim \mathcal{D}_\beta, \\ a' \sim \pi(\cdot | s')}} \left[ \left( Q_\theta(s, a) - r - \gamma Q_\theta(s', a') \right)^2 \right].$$

We follow [57, 53] and learn a target Q-network $Q_{\theta'}(s, a)$ in parallel for added stability. We use the AdamW algorithm [55] for optimizing the loss function in a minibatched setting, with gradient clipping applied to limit the norm of each gradient update to 1. The complete list of hyper-parameters used is provided in Table 7.

| Description | Hopper | Walker | HalfCheetah | Pendulum | Acrobot |
|---|---|---|---|---|---|
| number of hidden layers | 2 | 2 | 2 | 2 | 2 |
| number of neurons per layer | 500 | 500 | 500 | 256 | 100 |
| hidden activation function | sigmoid | sigmoid | sigmoid | sigmoid | sigmoid |
| learning rate of AdamW optimizer | 0.001 | 0.003 | 0.00003 | 0.003 | 0.001 |
| moving average of target $\theta'$ | 0.05 | 0.05 | 0.001 | 0.005 | 0.05 |
| training epochs (passes over data set) | 100 | 50 | 70 | 100 | 200 |
| batch size | 512 | 256 | 256 | 128 | 512 |

Table 7: Hyper-parameters for Fitted Q-Evaluation (FQE).

**Doubly Robust (DR).** [19, 37] leverages both importance sampling and value function estimation to construct a combined estimate that is accurate when either one of the individual estimates is correct. First, we define an estimate $\hat{Q}(s, a)$ of the Q-value function of policy $\pi$, and let $\hat{V}(s) = \mathbb{E}_{a \sim \pi(\cdot | s)}[\hat{Q}(s, a)]$ be the corresponding value estimate. We also define $\rho_t = \frac{\pi(a_t | s_t)}{\beta(a_t | s_t)}$ as the policy

ratio at step $t$. Then, the DR estimator is defined recursively as

$$V_{DR}^{t+1} = \hat{V}(s_t) + \rho_t \left( r_t + \gamma V_{DR}^t - \hat{Q}(s_t, a_t) \right),$$

such that the policy value estimate $\hat{J}_{DR}(\pi) = V_{DR}^0$. We parameterize both $\hat{Q}(s, a)$ and $\hat{V}(s)$ as MLPs and train them using AdamW in a mini-batched setting. Similar to FQE, we also update a target value network to improve convergence. The full list of hyper-parameters is provided in Table 8.

| Description | Hopper | Walker | HalfCheetah | Pendulum | Acrobot |
|---|---|---|---|---|---|
| number of hidden layers | 2 | 2 | 2 | 2 | 2 |
| number of neurons per layer | 500 | 500 | 500 | 256 | 100 |
| hidden activation function | sigmoid | sigmoid | sigmoid | sigmoid | sigmoid |
| learning rate of AdamW optimizer | 0.0003 | 0.003 | 0.003 | 0.003 | 0.00003 |
| moving average of target $\theta'$ | 0.05 | 0.05 | 0.05 | 0.05 | 0.001 |
| training epochs (passes over data set) | 50 | 50 | 50 | 100 | 100 |
| batch size | 32 | 256 | 512 | 256 | 128 |

Table 8: Hyper-parameters for Doubly Robust (DR) estimation.

**Importance Sampling (IS).** [33] evaluates the target policy by importance weighting the full trajectory returns in the behavior dataset, i.e.

$$\hat{J}_{IS}(\pi) = \mathbb{E}_{\tau \sim p_\beta} \left[ \left( \prod_{t=0}^{T-1} \frac{\pi(a_t|s_t)}{\beta(a_t|s_t)} \right) \sum_{t=0}^{T-1} \gamma^t R(s_t, a_t) \right].$$

It requires access to the target and behavior policy probabilities in order to compute the weighting. Specifically, we use the *per-decision* variant of IS (PDIS), i.e.

$$\hat{J}_{PDIS}(\pi) = \mathbb{E}_{\tau \sim p_\beta} \left[ \sum_{t=0}^{T-1} \gamma^t \left( \prod_{u=0}^{t} \frac{\pi(a_u|s_u)}{\beta(a_u|s_u)} \right) R(s_t, a_t) \right],$$

which has lower variance than IS.

**Density Ratio Estimation (DRE).** [31] estimates the ratio $w(s, a) = d^\pi(s, a)/d^\beta(s, a)$ of the discounted state-action occupancies of the target policy $\pi$ relative to the behavior policy $\beta$. The *discounted state-action occupancy* of a policy $\mu \in \{\beta, \pi\}$ is defined as

$$d^\mu(s, a) = \lim_{T \to \infty} \frac{\sum_{t=0}^{T} \gamma^t p(s_t = s, a_t = a \mid \mu)}{\sum_{t=0}^{T} \gamma^t},$$

where $p(s_t = s, a_t = a \mid \mu)$ indicates the probability of sampling state-action pair $(s, a)$ from $\mu$ at time step $t$. We also tested the variants of DICE [43] but found their performance to be unsatisfactory, so they have been omitted from the study. The target policy value is estimated as

$$\hat{J}(\pi) = \frac{1}{1 - \gamma} \mathbb{E}_{(s,a,r) \sim \mathcal{D}_\beta}[w(s, a) \cdot r].$$

$w(s, a)$ is parameterized as a feedforward neural network and its parameters are trained using Adam in a mini-batched setting. Fixed hyper-parameters necessary to reproduce the experiment are listed in Table 9. Additionally, since the method requires a kernel function to be specified, we use a Gaussian kernel $k(x, x') = \exp(-\eta\|x - x'\|^2)$, where $x$ and $x'$ are concatenations of the (standardized) state and action vectors. Since this requires setting a kernel bandwidth $\eta > 0$ which affects the overall performance significantly, we run this baseline for different values $\eta \in \{0.01, 0.1, 1, 10, 100\}$ and report the best performing result (according to log-RMSE).

The following model-based baseline methods were also chosen for empirical comparison with STITCH-OPE. They were chosen to determine the benefits of STITCH-OPE compared to fully autoregressive sampling, i.e. $w = 1$, and non-autoregressive sampling, i.e. $w = T$.

**Model-Based (MB).** [19, 40] consists of learning dynamics $\hat{P}(s'|s, a)$, reward function $\hat{R}(s, a)$ and termination function $\hat{D}(s)$ trained on the behavior dataset to directly approximate the data-generating

| Description | Value |
|---|---|
| number of hidden layers of $w(s,a)$ | 2 |
| number of neurons per layer of $w(s,a)$ | 256 |
| hidden activation function | Leaky ReLU |
| output activation function | SoftPlus |
| learning rate of Adam optimizer | 0.001 |
| training epochs (passes over the data set) | 20 (D4RL), 200 (Gym) |
| batch size | 512 |

Table 9: Hyper-parameters for Density Ratio Estimation (DRE) [31].

distribution of the target policy, $p_\pi(\tau)$. $\hat{P}$ directly predicts the next state $s'$ given the current state $s$ and action $a$. Both $\hat{P}$ and $\hat{R}$ can be found by solving a standard nonlinear regression problem, and $\hat{D}$ can be found by solving a binary classification problem trained on termination flags in the behavior dataset. We parameterize all functions as nonlinear MLPs and obtain their optimal parameters using Adam in a mini-batched setting. Once we obtain their optimal parameters, we estimate the target policy return by generating 50 length-$T$ rollouts from the estimated model, and average their empirical cumulative returns. The necessary hyper-parameters are described in Table 10.

| Description | Value |
|---|---|
| number of hidden layers | 3 |
| number of neurons per layer | 500 |
| hidden activation function | ReLU |
| learning rate of Adam optimizer | 0.0003 |
| training epochs (passes over data set) | 100 |
| batch size | 1024 |

Table 10: Hyper-parameters for Model-Based (MB) estimation.

**Policy-Guided Diffusion (PGD).** [16] takes a generative approach by simulating target policy trajectories using a guided diffusion model. We follow the original implementation by training a diffusion model on the behavior data, using the official implementation located at `https://github.com/EmptyJackson/policy-guided-diffusion` (MIT licensed). We then generate 50 full-length trajectories from the model using guided diffusion [18] with the guidance function $g_{simple}(\tau) = \nabla_\tau \sum_t \log \pi(a_t|s_t)$, using which we estimate the empirical return of the target policy. All hyper-parameters for training the diffusion models are fixed as per the original paper and codebase (see Appendix A therein for details). However, we found that the policy guidance coefficient $\alpha$ and guidance normalization both have significant effects on performance, thus we ran PGD for different choices of $\alpha \in \{0.001, 0.01, 0.1, 1.0, 10, 100, 1000\}$ with and without guidance normalization, and report the best performing result (according to log-RMSE).

## I  Metrics

Let $\pi_1, \ldots \pi_{10}$ be the target policies, $\hat{J}_1(\pi_i), \hat{J}_2(\pi_i), \ldots \hat{J}_5(\pi_i)$ be the estimates of the target policy values across the 5 seeds, and $J(\pi_1), \ldots J(\pi_{10})$ be the target policy values estimated using 300 rollouts collected by running the target policies in the environments.

The following metrics were used to quantify and compare the performance of STITCH-OPE and all metrics:

**Log Root Mean Squared Error (LogRMSE).** This is defined as the log root mean squared error using the estimates $\hat{J}_j(\pi_1), \ldots \hat{J}_j(\pi_{10})$ and the ground truth returns $J(\pi_1), \ldots \ldots J(\pi_{10})$, averaged across seeds $j = 1 \ldots 5$. Mathematically,

$$\frac{1}{5} \sum_{j=1}^{5} \log \sqrt{\frac{1}{10} \sum_{i=1}^{10} (\hat{J}_j(\pi_i) - J(\pi_i))^2}.$$

**Spearman (Rank) Correlation.** This is defined as the Spearman correlation [62] between the estimates $\hat{J}_j(\pi_1), \ldots \hat{J}_j(\pi_{10})$ and the ground truth returns $J(\pi_1), \ldots \ldots J(\pi_{10})$, averaged across seeds $j = 1 \ldots 5$.

**Regret@1.** This is defined as the absolute difference in return between the best policy selected using the baseline policy returns $\hat{J}_j(\pi_i)$ and the policy selected according to the ground truth estimates $J(\pi_i)$, averaged across seeds $j = 1 \ldots 5$, i.e:

$$\frac{1}{5} \sum_{j=1}^{5} \left| J(\pi_{i_j^{max}}) - \max_{i=1\ldots10} J(\pi_i) \right|, \qquad \text{where} \quad i_j^{max} = \text{argmax}_{i=1\ldots10} \hat{J}_j(\pi_i).$$

**Normalization.** In order to compare metrics consistently across environments, we follow [14] and use the normalized policy values:

$$\frac{\hat{J}_j(\pi_i) - V_{min}}{V_{max} - V_{min}}, \qquad \text{where} \quad V_{min} = \min_i J(\pi_i), \quad V_{max} = \max_i J(\pi_i),$$

where $V_{min}$ and $V_{max}$ are the minimum and maximum target policy values, respectively.

**Error Bars.** All tables and figures report error bars defined as +/- one standard error, i.e. $\hat{\sigma}/\sqrt{n}$ where $\hat{\sigma}$ is the empirical standard deviation of each metric value across seeds and $n$ is the number of seeds (fixed to 5 for all experiments).

## J STITCH-OPE Training and Hyper-Parameter Details

**Diffusion Model Architecture.** We follow the configuration used in [18] for training the diffusion model, including architecture, optimizer, and noise schedule. Specifically, we parameterize the diffusion process $\epsilon$ as a UNet architecture with residual connections [61], trained with a cosine learning rate schedule [56]. The UNet contains 6 repeated residual blocks, each consisting of two temporal convolutions followed by group norm and a final Mish nonlinearity. The time step embedding is produced by a single fully connected layer and added to the activations of the first temporal convolution within each block.

**Diffusion Model Training Hyper-Parameters.** The list of training hyper-parameters for the trajectory diffusion model is provided in Table 11.

| Description | Value |
|:---:|:---:|
| diffusion architecture | UNet |
| denoising time steps | 256 |
| learning rate of Adam optimizer | 0.0003 |
| training epochs (passes over the data set) | 150 |
| batch size | 128 |
| training steps per epoch | 5000 (D4RL), 2000 (Gym) |
| guidance coefficient for $\pi$, i.e. $\alpha$ | 0.5 (D4RL), 0.1 (Gym) |
| guidance coefficient ratio for $\beta$, i.e. $\frac{\lambda}{\alpha}$ | 0.5 (D4RL), 1 (Gym) |
| window size of sub-trajectories, i.e. $w$ | 8 (D4RL), 16 (Gym) |

Table 11: Hyper-parameters for STITCH-OPE.

**Reward Function.** The reward function approximation $\hat{R}(s, a)$ is a two-layer MLP with ReLU activations and 32 neurons per hidden layer, and is trained using Adam with a learning rate of $0.001$ and batch size of $64$.

**Guidance Coefficients.** For Gym domains, we use $\alpha = \lambda = 1$, corresponding to the theoretically justified guidance function in Equation 8, assuming low distribution shift. For D4RL tasks, we use tempered values $\alpha = 0.5$ and $\lambda = 0.25$ to improve sample stability and regularization, which we found empirically helpful in higher-dimensional settings.

**Sub-Trajectory Length.** We use $w = 16$ for Gym domains and $w = 8$ for most D4RL tasks. For HalfCheetah, we reduce to $w = 4$ due to the environment's fast dynamics, which caused degradation

in stitching fidelity with longer sub-trajectories. Also to add stability for cheetah, we set the clip denoised flag to True during the backward diffusion process.

## K    Diffusion Policy Training and Evaluation

**Diffusion Policy Loss.** We follow [41] and parameterize each target policy $\pi'_i$, $i = 1 \ldots 10$ as a conditional diffusion model $\epsilon_{\phi_i}(a^k, k|s)$, whose parameters $\phi_i$ are learned by optimizing the behavior cloning objective (compare with (1))

$$\mathcal{L}(\phi_i) = \mathbb{E}_{k,\,\epsilon \sim \mathcal{N}(0,I),\,s \sim \mathcal{D}_\beta,\,a \sim \pi_i(\cdot|s)} \left[ \|\epsilon - \epsilon_{\phi_i}(a^k, k|s)\|^2 \right].$$

**Diffusion Policy Guidance Score.** In order to use the fine-tuned $\epsilon_{\phi_i}(a^k, k|s)$ as a guidance function for off-policy evaluation in STITCH-OPE, we use the following equivalence between score-based models and denoising diffusion [9] (extended trivially to the conditional setting)

$$\nabla_a \log \pi'_i(a|s)|_{a=a^k} = -\frac{\epsilon_{\phi_i}(a^k, k|s)}{\sigma_k}.$$

Specifically, this expression cannot be calculated at $k = 0$ since $\sigma_0 = 0$ using the standard parameterization of diffusion models, so we approximate it at $k = 1$ and use the resulting gradient in STITCH-OPE.

**Implementation and Hyper-Parameters.** We implement the diffusion model using the Clean-Diffuser package [50] with official repository at `https://github.com/CleanDiffuserTeam/CleanDiffuser` (Apache 2.0 licensed). To train the diffusion policies, we first generate rollouts from each of the pre-trained target policies in D4RL [14], and then minimize the behavior cloning objective $\mathcal{L}(\phi_i)$ above to obtain the diffusion policy parameters. The list of relevant hyper-parameters is provided in Table 12.

| Description | Value |
|---|---|
| embedding dimension | 64 |
| hidden layer dimension | 256 |
| learning rate | 0.0003 |
| diffusion time steps | 32 |
| EMA rate | 0.9999 |
| total training steps | 10000 |
| number of transitions to generate for each dataset | 1000000 |
| training batch size | 256 |

Table 12: Hyper-parameters for training diffusion policies.

## L    Additional Experiments

### L.1    Sensitivity to Guidance Coefficients $\alpha$ and $\lambda$

We evaluate STITCH-OPE across different choices of the guidance coefficients $\alpha$ and $\lambda$, and plot the resulting trends in Figure 5 for Hopper and Figure 6 for Walker2D. Each plot is generated by applying bicubic interpolation to the grid evaluations of the Spearman correlation and LogRMSE. The optimal coefficient values of $\alpha$ and $\lambda$ remain consistent across environments. The optimal balance for off-policy evaluation is attained by assigning a moderate coefficient for the target policy score $\alpha$ (i.e. $\alpha < 1$) and a smaller but positive coefficient to the behavior policy score $\lambda$, i.e. $0 < \lambda < \alpha$. Recall that $\lambda$ controls the amount of distribution shift we are willing to accept during guided trajectory generation. $\lambda = 1$ is theoretically unbiased, but potentially under-regularized and leads to dynamically infeasible (high-variance) samples. Meanwhile, $\lambda = 0$ is often over-regularized and leads to trajectories that are heavily biased towards the behavior policy $p_\beta(\tau)$. From the plots, we see that a moderate amount of regularization is optimal (around 25% of the value of $\alpha$), which is consistent with regularization in supervised machine learning (i.e., regression).

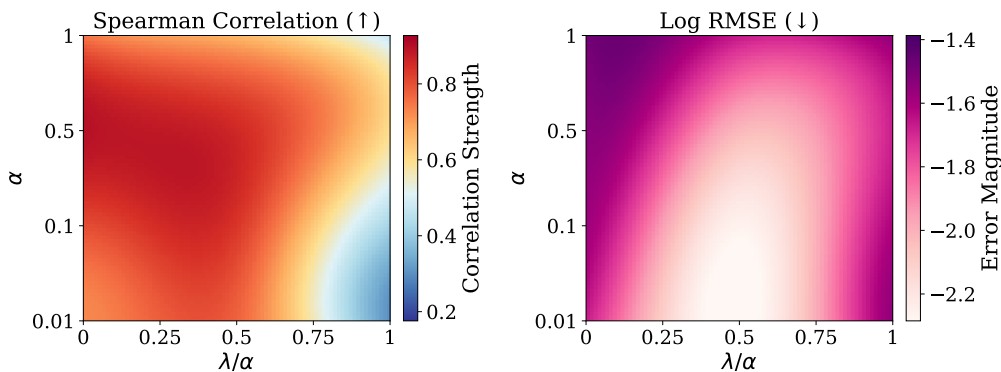

Figure 5: Smoothed performance landscape for Hopper. **Left:** Spearman correlation is largest around $\alpha \in [0.1, 0.5]$, $\lambda \le 0.5\alpha$. **Right:** The LogRMSE is smallest around $\alpha \in [0.01, 0.5]$, $\lambda \in [0.25\alpha, 0.75\alpha]$. These results confirm the optimal range of $\lambda$ is $0 < \lambda < \alpha$.

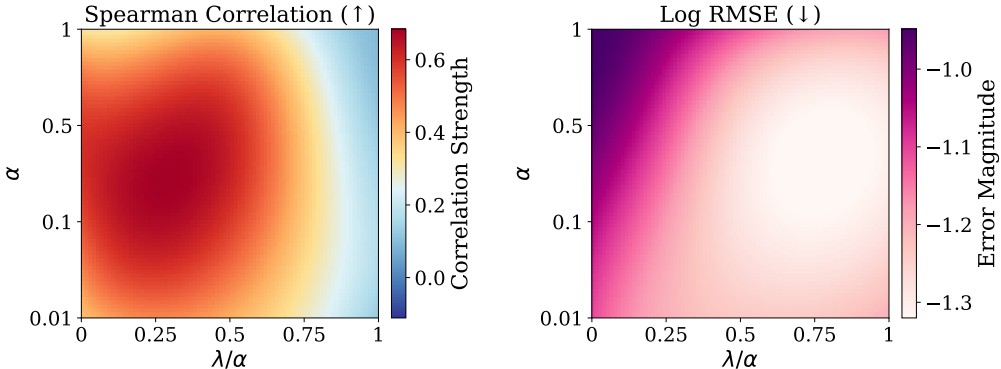

Figure 6: Smoothed performance landscape for Walker2d. Results are generally consistent with Hopper. **Left:** Spearman correlation is largest around $\alpha \in [0.1, 0.5]$, $\lambda \approx 0.25\alpha$. **Right:** The LogRMSE is smallest around $\alpha \in [0.1, 0.5]$, $\lambda \approx 0.75\alpha$. These results confirm the optimal range of $\lambda$ is $0 < \lambda < \alpha$.

## L.2 Sensitivity to Window Size $w$

To further analyze the sensitivity to $w$, we evaluate STITCH-OPE across different intermediate values of $w$, and compare the performance according to LogRMSE and Spearman correlation metrics. As illustrated in Figure 7 for Hopper and 8 for Pendulum, the best performance is consistently achieved using moderate values of $w$, i.e. $w = 8$ for Hopper and $w = 16$ for Pendulum. As hypothesized in the main text, based on our analysis in Section 3.3 and Section 3.4, low values of $w$ provide more flexibility when stitching trajectories and thus promote compositionality, but are more susceptible to the compounding of errors. High values of $w$ are less susceptible to error compounding but at the expense of compositionality and thus less adaptability to distribution shift. In the current ablation experiment, it is clear that the best balance between compositionality and error compounding occurs using moderate values of $w$, and the greatest deterioration in performance occurs for very small or very large values. It is also important to note that increasing $w$ reduces inference speed due to longer trajectory generations per diffusion step, highlighting a practical trade-off between computational cost and evaluation accuracy.

## L.3 Sensitivity to the Choice of Behavior Policy/Behavior Dataset

To examine the sensitivity of STITCH-OPE to the choice of behavior policy or behavior dataset, we ran our algorithm and all baselines on the medium-expert and expert data sets for the Hopper environment. Across all reported results, we set the hyper-parameters to $T = 768$ and $\gamma = 0.99$, which allows for easy comparison between the data sets. The results for the medium-expert dataset are

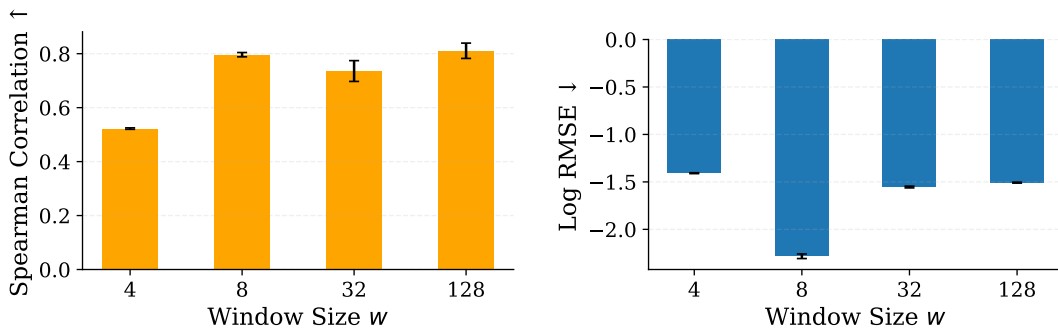

Figure 7: Sensitivity of STITCH-OPE to window size $w$ in the Hopper-v2 environment. **Left:** Spearman rank correlation. **Right:** Log RMSE. Error bars denote one standard error over five random seeds. The overall best performance is attained for $w = 8$, suggesting a good balance between compositionality and error compounding.

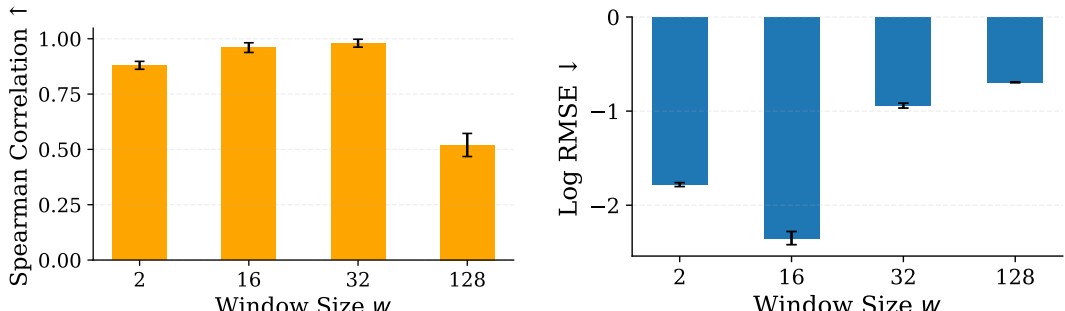

Figure 8: Sensitivity of STITCH-OPE to window size $w$ in the Pendulum-v1 environment. **Left:** Spearman rank correlation. **Right:** Log RMSE. Error bars denote one standard error over five random seeds. The overall best performance is attained for $w = 16$, suggesting a good balance between compositionality and error compounding.

shown in Table 13, and the results for the expert dataset are shown in Table 14. STITCH-OPE soundly outperforms all other baselines in 5 out of 6 instances (metric-dataset combinations). Furthermore, while the performance of some baseline methods exhibits significant variation between data sets (i.e., FQE, DR, IS, PGD in Log RMSE), the performance of STITCH-OPE remains the most consistent across all metrics and datasets. Finally, we observe a steep decline in performance of the baselines on the medium-expert and expert data sets as compared to the medium data set, which suggests that prior OPE work is not robust to large distribution shifts between the behavior and target policies.

| Metric | FQE | DR | IS | DRE | MB | PGD | **Ours** |
|---|---|---|---|---|---|---|---|
| Log RMSE ↓ | -1.14 ± 0.03 | -0.71 ± 0.05 | -0.15 ± <0.01 | -0.18 ± <0.01 | -1.33 ± <0.01 | -0.55 ± 0.02 | **-2.17 ± 0.01** |
| Rank Corr. ↑ | 0.51 ± 0.09 | 0.22 ± 0.15 | 0.11 ± 0.06 | 0.13 ± 0.14 | 0.75 ± 0.01 | 0.77 ± 0.04 | **0.83 ± <0.01** |
| Regret@1 ↓ | 0.02 ± <0.01 | 0.06 ± 0.04 | 0.60 ± 0.15 | 0.13 ± 0.05 | **0.00 ± 0.00** | 0.01 ± <0.01 | **0.00 ± 0.00** |

Table 13: Comparison of OPE methods for Hopper-medium-expert. Error bars represent ± one standard error across 5 seeds; any value shown as <0.01 is nonzero but rounds to zero at two decimals.

| Metric | FQE | DR | IS | DRE | MB | PGD | **Ours** |
|---|---|---|---|---|---|---|---|
| Log RMSE ↓ | -0.39 ± <0.01 | 0.02 ± <0.01 | -1.05 ± <0.01 | 0.02 ± <0.01 | -1.46 ± <0.01 | -0.01 ± <0.01 | **-2.33 ± 0.01** |
| Rank Corr. ↑ | 0.46 ± 0.03 | 0.20 ± 0.11 | 0.26 ± 0.1 | 0.20 ± 0.11 | 0.28 ± <0.01 | 0.47 ± 0.03 | **0.51 ± 0.04** |
| Regret@1 ↓ | 0.12 ± 0.04 | 0.1 ± 0.04 | 0.32 ± 0.07 | 0.1 ± 0.04 | 0.18 ± <0.01 | **0.00 ± 0.00** | 0.04 ± 0.01 |

Table 14: Comparison of OPE methods for Hopper-expert. Error bars represent ± one standard error across 5 seeds; any value shown as <0.01 is nonzero but rounds to zero at two decimals.

## L.4  Sensitivity to Reward Estimation

Recall that the immediate reward function in STITCH-OPE is modeled by a neural network and trained on the offline transitions $(s, a, r, s', ...)$, which can be described as a standard (nonlinear) regression problem. We evaluate the sensitivity of STITCH-OPE to this modeling choice by running STITCH-OPE given the true (unknown) reward function, and comparing this to the original results on the estimated reward function. Table 15 shows the aggregated metrics and the per-target-policy results on the Hopper-medium benchmark dataset. In summary, the accumulated errors from reward estimation have a minimal effect on OPE performance as measured using aggregate metrics, and the error remains stable and low across all target policies.

| Metric | True Reward | Estimated Reward |
|---|---|---|
| Log RMSE $\downarrow$ | -2.30 ± 0.01 | -2.28 ± 0.01 |
| Rank Corr. $\uparrow$ | 0.77 ± + 0.01 | 0.75 ± 0.01 |
| Regret@1 $\downarrow$ | 0.10 ± 0.03 | 0.13 ± 0.03 |

Table 15: Aggregated metrics obtained by running STITCH-OPE given the true reward function and the estimated reward function on Hopper-medium. Error bars represent ± one standard error across 5 seeds.

| Policy | 1 | 2 | 3 | 4 | 5 | 6 | 7 | 8 | 9 | 10 |
|---|---|---|---|---|---|---|---|---|---|---|
| True Reward | 137.52 | 229.49 | 218.71 | 233.40 | 268.40 | 255.46 | 258.95 | 239.99 | 260.88 | 268.18 |
| Est. Reward | 135.43 | 228.80 | 217.35 | 229.55 | 266.77 | 253.43 | 253.87 | 235.85 | 256.79 | 264.88 |
| Rel. Error % | -1.52 | -0.30 | -0.62 | -1.65 | -0.61 | -0.79 | -1.96 | -1.72 | -1.57 | -1.23 |

Table 16: Per-target-policy values obtained by running STITCH-OPE given the true reward function and the estimated reward function on Hopper-medium.

## L.5  Trajectory Visualizations

We visualize and compare trajectories generated by the guided and unguided versions of STITCH-OPE and Policy-Guided Diffusion (PGD) [16] against both random and optimal policies. These visualizations highlight differences in the quality of generated trajectories, alignment with target policies, and generalization capabilities across various environments. As shown in Figures 9 and 11, STITCH-OPE closely mimics the target policy behavior. On the other hand, PGD performs poorly, significantly overestimating the performance of the random policy. Figure 10 further demonstrates that STITCH-OPE maintains consistent and robust behavior across policy settings.

## L.6  Comparison with Diffusion World Models

The Diffusion World Model (DWM) [10] is an alternative approach that leverages diffusion models for trajectory generation. However, DWMs are not well-suited for the off-policy evaluation problem, as we demonstrate experimentally in Table 17. In particular, DWM generates only future states and rewards, e.g., $p(s_{1:T}, r_{1:T} \mid s_0, a_0, g)$, discarding most action information. Consequently, it provides no mechanism for policy guidance and cannot adapt to the significant distribution shift present between the behavior and target policies.

By contrast, STITCH explicitly models the joint state–action trajectory $p(\tau) = p(s_0, a_0, \ldots, s_T)$; we therefore effectively mitigate action distribution shift by incorporating policy guidance into trajectory generation (Eq. 3), which steers $a_{0:T}$ toward the target policy. In Markov settings, the DWM diffusion step for $s_t$ depends primarily on $s_{t-1}$ and marginalizes over the behavior policy at $t - 1$, so rollouts remain near the behavior occupancy and are harder to steer, making DWM more brittle under action distribution shift.

More generally, while there have been significant advancements in diffusion modeling over the last few years, policy guided diffusion (PGD) [16] is the most representative and recent model-based baseline found at the time of this publication which is relevant for OPE. This is because addressing distribution mismatch between behavior and target policy trajectories is a highly non-trivial problem,

which only STITCH-OPE (and partially PGD) are able to tackle effectively. For a fuller discussion of diffusion-based baselines and other model-based OPE methods, we refer the reader to Section N.

| Metric | DWM | PGD | **Ours** |
|---|---|---|---|
| Log RMSE ↓ | -0.82 ± 0.20 | -1.22 ± 0.02 | **-2.33 ± 0.02** |
| Rank Corr. ↑ | 0.41 ± 0.10 | 0.36 ± 0.09 | **0.76 ± 0.02** |
| Regret@1 ↓ | 0.14 ± 0.11 | **0.04 ± 0.01** | 0.11 ± 0.04 |

Table 17: Comparison of diffusion modeling approaches on the Hopper-medium benchmark. Error bars represent ± one standard error across 5 seeds.

# M  Computing Resources

**Hardware and Software.** All experiments were conducted on a local workstation running Ubuntu 20.04 LTS and Python 3.9, with the following hardware:

- 2× NVIDIA RTX 3090 GPUs (24 GB each)

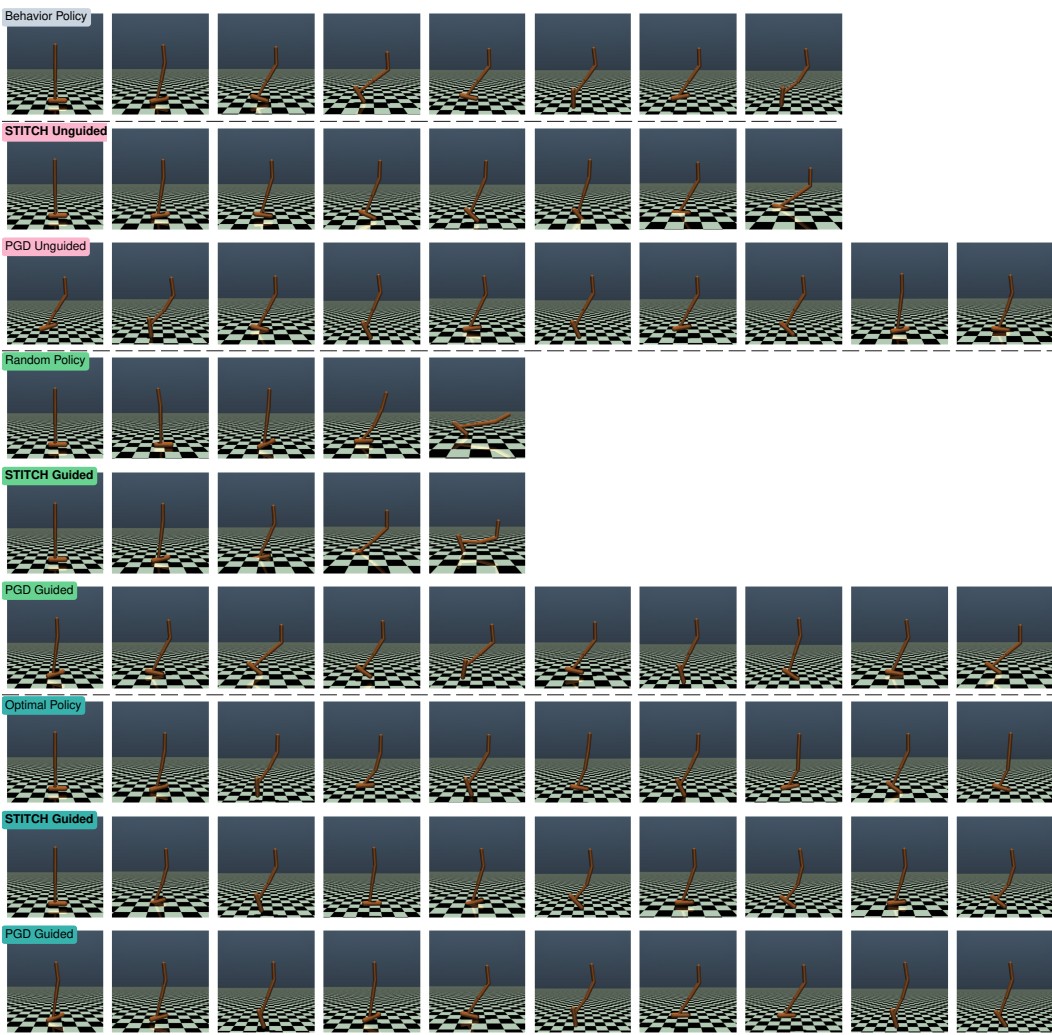

Figure 9: Trajectory visualizations in the Hopper environment. Both STITCH-OPE and PGD track the optimal policy. PGD significantly overestimates the performance of the random policy, while STITCH-OPE correctly models both the state trajectory and the termination.

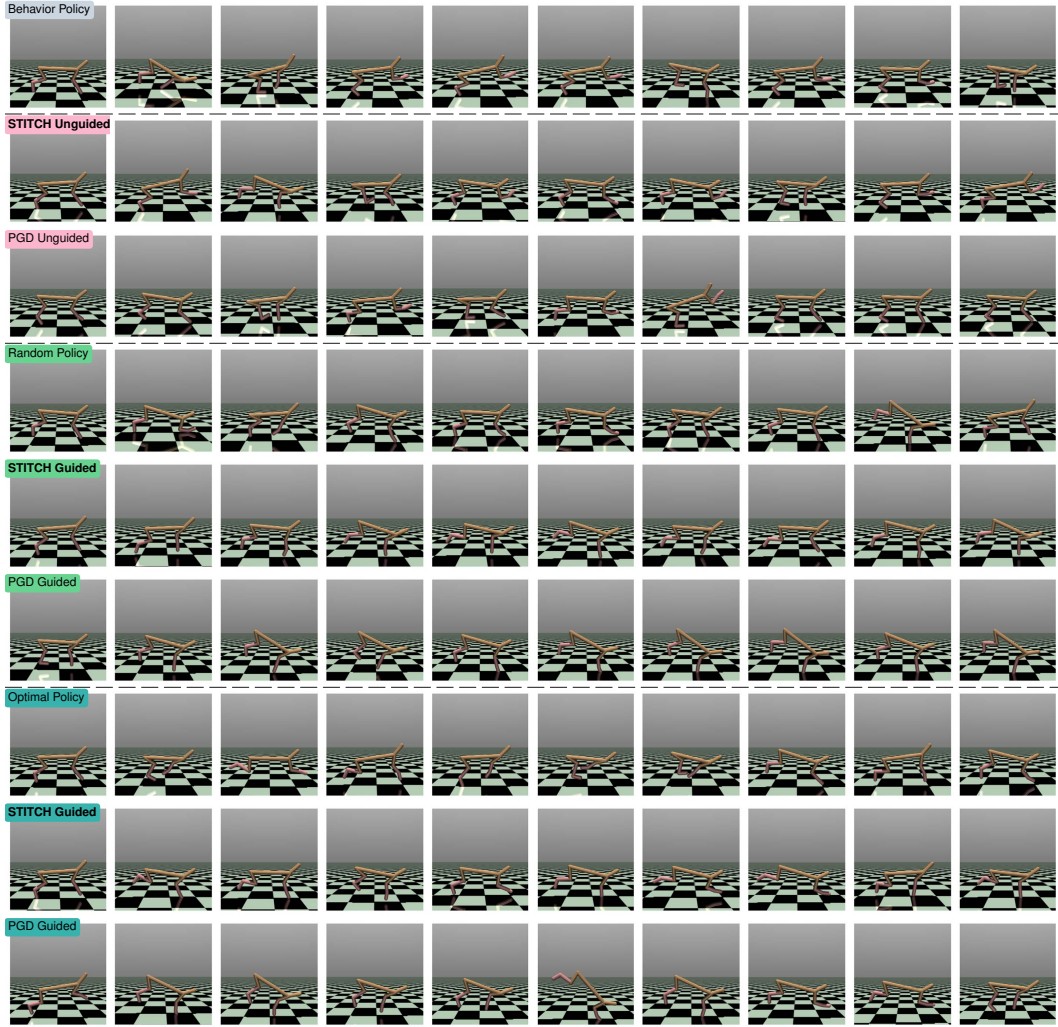

Figure 10: Trajectory visualizations in the HalfCheetah environment. STITCH-OPE and PGD both demonstrate consistent behavior across all policy types, highlighting their robust generalization on this task.

- Intel(R) Core(TM) i9-9820X CPU @ 3.30GHz (10 cores / 20 threads)
- 128 GB RAM.

**Runtime.** Each full training of a diffusion model for a D4RL task took approximately 20 hours to complete, depending on environment complexity and rollout length. Each OpenAI Gym task took approximately 5 hours. Each evaluation for a D4RL environment took around 18 hours in total (across all 5 seeds) to complete, and each OpenAI Gym environment took around 6 hours to complete.

# N Related Work

Off-policy evaluation plays a critical role in offline reinforcement learning, enabling the evaluation of policies without directly interacting with the environment. OPE has been studied across a wide range of different domains including robotics [52], healthcare [58, 60, 59] and recommender systems [51, 63]. Relevant work includes model-free and model-based OPE approaches, including recent generative methods in offline RL.

**Model-Free Methods.** Model-free methods, such as Importance Sampling (IS) and per-decision Importance Sampling (PDIS) [33] reweight trajectories (or single-step transitions) from the behavior

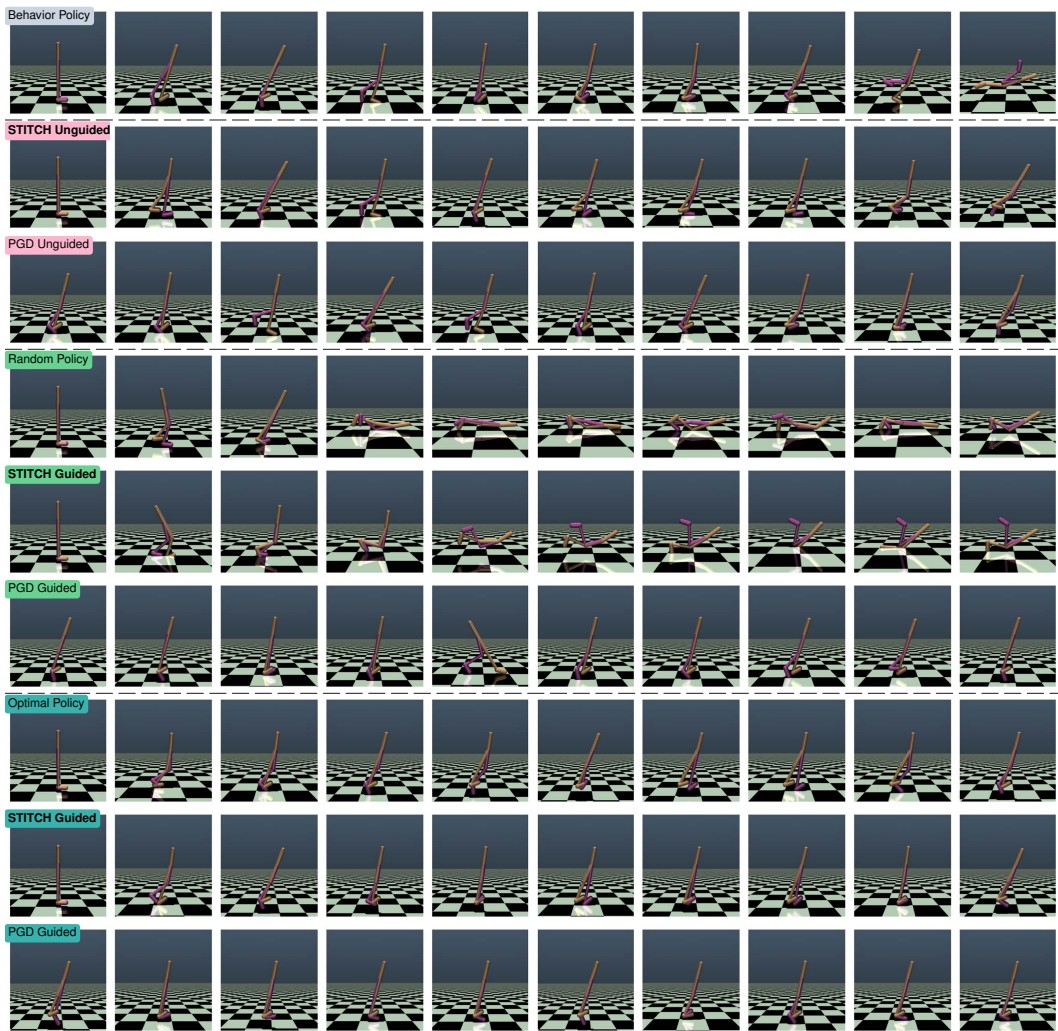

Figure 11: Trajectory visualizations in the Walker2d environment. STITCH-OPE effectively imitates both random and optimal policies. As for the Hopper environment, PGD struggles to correctly imitate the random policy, significantly overestimating its performance.

policy to approximate returns under a target policy. However, this class of methods suffers from the so-called "curse of horizon", in which the variance grows exponentially in the length of the trajectory [25, 27]. Doubly Robust (DR) methods [19, 37, 11] further combine estimation of value functions with importance weights, reducing the overall variance. Distribution-correction methods (DICE) [32, 43, 47] and their variants [25, 31] try to mitigate the curse-of-horizon by performing importance sampling from the stationary distribution of the underlying MDP. However, these methods perform relatively poorly on high-dimensional long-horizon tasks [14].

**Model-Based Methods.** Model-based OPE methods estimate the target policy value by learning approximate transition and reward models from offline data and simulating trajectories under the target policy [19, 21]. These methods have shown strong empirical performance, especially in continuous control domains [38, 46], but they often suffer from compounding errors during rollouts, which can lead to biased estimates in high-dimensional or long-horizon settings [20, 17].

**Offline Diffusion.** Inspired by the recent performance of diffusion models across many areas of machine learning [15, 9], a new stream of reinforcement learning has emerged which leverages diffusion models trained on behavior data [29, 48]. Janner et al. [18], Ajay et al. [1] train diffusion models on behavior data that can be guided to achieve new goals. Ding et al. [10] uses diffusion models as world models. While successful at modeling the behavior distribution over trajectories

accurately, diffusion world models do not use guidance and thus cannot address the policy distribution shift. Jackson et al. [16], Rigter et al. [35] apply guided diffusion to offline policy optimization by setting the guidance function to be the score of the learned policy, while [64] applies guided diffusion to satisfy added safety constraints. Unlike STITCH-OPE, these works do not use negative guidance nor stitching, which we found leads to unstable policy values when applied directly for offline policy evaluation over a long-horizon. Mao et al. [30] applies DICE to estimate the stationary distribution of the underlying MDP, which is used as a guidance function to correct the policy distribution shift for offline policy optimization. Unlike STITCH-OPE, this work is not directly applicable to offline policy evaluation. Finally, Li et al. [54] introduces a variant of trajectory stitching for augmenting behavior data, but does not apply it for offline policy evaluation. To the best of our knowledge, STITCH-OPE is the first work to apply diffusion models to evaluate policies on offline data.

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
