# OpenReview forum: "STITCH-OPE: Trajectory Stitching with Guided Diffusion for Off-Policy Evaluation"
_NeurIPS.cc/2025/Conference — NeurIPS 2025 spotlight_

### Official Review · Reviewer_fKxK · 2025-06-05

**Clarity:** 2
**Significance:** 3
**Originality:** 3
**Rating:** 4
**Confidence:** 3

**Summary:**

This paper proposes a method called Stitch-OPE for evaluating the performance of policies trained on offline data. Stitch-OPE first trains a diffusion model capable of generating sub-trajectories based on data collected by the behavior policy. It then estimates the distribution of the target policy using importance sampling. Finally, it evaluates the performance of trajectories generated under the target policy using a learned reward predictor. This work addresses a key challenge in offline reinforcement learning. However, it may suffer from certain inaccuracies in out-of-distribution (OOD) regions.

**Questions:**

1. What is the direct relationship between Figure 1 and Off-Policy Evaluation?

2. How does Stitch-OPE handle the OOD (out-of-distribution) issue? Intuitively, since Stitch-OPE trains a reward predictor using behavior data, wouldn't it lead to significant errors in evaluating the performance of the target policy if the trajectories generated by the target policy fall into OOD regions?

**Ethical Concerns:**

["NO or VERY MINOR ethics concerns only"]

**Final Justification:**

The authors' response has addressed most of my concerns.

**Limitations:**

This method may suffer from certain inaccuracies in out-of-distribution (OOD) regions.

**Quality:**

3

**Strengths And Weaknesses:**

### Strengths
1. The experimental results are relatively strong.
2. The paper provides detailed experimental settings.

### Weaknesses
1. The baselines in Figure 2 and Table 2 are inconsistent. Figure 2 labels the baseline as "MBR", while Table 2 uses "MB".
2. The content illustrated in Figure 1 does not seem to be directly related to Off-Policy Evaluation.

---

> ### Author Rebuttal · Authors · 2025-07-31
>
> We would like to thank the reviewer for their detailed comments. We appreciate that the reviewer found our empirical results versus a variety of recent model-based and model-free OPE methods convincing, including the detailed ablation studies testing various hyper-parameter choices of STITCH-OPE. Please find our responses to your questions below.
>
> ---
>
> ## (Q1) What is the direct relationship between Figure 1 and Off-Policy Evaluation?
>
> Figure 1 illustrates that several components and innovations in model-based reasoning are required to achieve robust model-based OPE in practice. This includes:
> 1. careful processing of the offline data set to ensure good sample efficiency
> 2. chunked prediction, composition and guidance during inference are crucial to handle policy distribution shift, which is central to all OPE methods, and
> 3. empirical reward and return estimation, which is the main task in OPE.
>
> In more details:
> 1. Figure 1 (Part A) shows how the offline data is split into overlapping sub-trajectories. Critical to model-based OPE methods, this builds an offline training data set on which accurate short-horizon transition dynamics can be learned using less offline data.
> 2. Once these dynamics are learned using a chunked diffusion model, we effectively obtain a model that accurately captures how the state evolves under the behavior policy (Part B). A critical question of all OPE methods is how to tackle the distribution shift that arises between the behavior and target policy. Figure 1 (Part D) shows that this question can be effectively addressed by sub-trajectory composition and target policy guidance. These ideas work synergistically to account for the policy distribution shift, thus accurately approximating what would happen had we executed the target policy in the real world, without access to a simulator or the real world which is a key feature of OPE.
> 3. Finally, return estimation involves estimating the empirical reward model from the offline data (Part C), and together with accurate synthetic rollouts from our diffusion model, we can generate accurate estimates of the target policy value, which is the main goal of OPE.
>
> ---
>
> ## (Q2) How does Stitch-OPE handle the OOD (out-of-distribution) issue? Intuitively, since Stitch-OPE trains a reward predictor using behavior data, wouldn't it lead to significant errors in evaluating the performance of the target policy if the trajectories generated by the target policy fall into OOD regions?
>
> Great question! We thank the reviewer for raising a very important point about the necessary assumptions for successful implementation and the limitations of STITCH-OPE.
>
> In summary, we expect STITCH-OPE to generate novel trajectories, but by leveraging the existing structure of the behavior data set through correct composition and stitching of the data, we believe the burden of predicting reward on OOD transitions is significantly reduced because stitching and composition do not involve generating OOD states. We now explain this point in greater detail:
>
> **Clarification on the Meaning of OOD:** When we refer to out-of-distribution in the context of STITCH or other OPE methods, we mean that rollouts simulated from behavior and target policies might be extremely dissimilar (i.e. in terms of standard measure like L2 norm, KL-divergence), but at the level of individual transitions or chunks, they must still share some common structure. This is typically expressed in OPE literature as requiring that the support of the target policy's action distribution (at every individual decision time step) must be within the support of the behavior distribution, which is also required by STITCH-OPE (Section 3.4). Consequentially, in the normal application of STITCH-OPE, we *do* expect to generate entirely novel trajectories through effective composition and stitching, but individual transitions within these rollouts are still expected be connected to the behavior data.
>
> **Stitching Improves Reward Estimation in OOD Regions:**
> We also argue that successful stitching is a necessary component of accurate reward estimation, since it allows the synthetic rollouts to better match the target rollout distribution, while also respecting the overlapping policy support assumption that is necessary for successful reward prediction (for instance, see row A of Table 1 in our paper).
>
> **Standard Model-Based RL Assumptions:** We agree with the reviewer that estimation of immediate reward can be particularly challenging when it is nonsmooth or non-Lipschitz, i.e. discontinuous or pathological, even with successful stitching. However, our practical assumption is that the reward is both Markovian and sufficiently well-behaved that it can be approximated from the offline data sets in standard RL benchmarks. This assumption also underlies all model-based offline RL methods that leverage deep learning to approximate both the dynamics and the reward function for value estimation.
>
> **Accurate Dynamics Estimation is a Difficult Problem in RL:** We also want to emphasize that learning the state dynamics accurately from purely offline data -- while also avoiding the error compounding problem over long horizons during evaluation -- is very often highly non-trivial in OPE. For instance, even if errors of estimated $P(s' | s, a)$ are moderately small at any given time step, they can quickly blow up over time rendering them ineffective over sufficiently long horizon. Meanwhile, reward estimation error typically does not compound, since rewards are predicted individually at the transition level and thus their error accumulates linearly in $T$. The recent paper of Jiang [1] discusses a number of practical challenges of dynamics estimation that are not present in reward estimation, including error compounding and challenges of accurately learning stochastic dynamics. It also shows generally that the error in value increases quadratically in terms of the dynamics error; this is asymptotically looser than the bound on the value error that grows linearly in terms of the reward error. Thus, we wish to re-emphasize that the main goal of STITCH-OPE is to estimate the dynamics accurately offline through composition and stitching, which tackles a crucial problem in model-based OPE that has been inadequately addressed by prior work.
>
> **Ablation Experiment to Study the Effects of Reward Estimation on D4RL:** To determine the practical effect of reward estimation on the performance of STITCH-OPE on standard offline benchmarks, we ran STITCH-OPE on the Hopper-medium data set using both the true reward function and the empirically estimated reward function. The aggregated performance comparison is provided below:
>
> Aggregated Reward Ablation on Hopper-medium
> | Metric | True Reward | Estimated Reward |
> | :--- | :-- | :--- |
> | Log-RMSE | -2.30 ± 0.01| -2.28 ± 0.01 |
> | Rank Correlation | 0.77 ± + 0.01  | 0.75 ± 0.01|
> | Regret@1 | 0.10 ± 0.03  | 0.13 ± 0.03 |
>
> and the per-policy performance is also included:
>
> Per-Target-Policy Reward Ablation on Hopper-medium
> | Target Policy | 0 | 1 | 2 | 3 | 4 | 5 | 6 | 7 | 8 | 9 |
> | :--- | :--- | :--- | :--- | :--- | :--- | :--- | :--- | :--- | :--- | :--- |
> | Value using True Reward | 137.52 | 229.49 | 218.71 | 233.40 | 268.40 | 255.46 | 258.95 | 239.99 | 260.88 | 268.18 |
> | Value using Est. Reward | 135.43 | 228.80 | 217.35 | 229.55 | 266.77 | 253.43 | 253.87 | 235.85 | 256.79 | 264.88 |
> | Relative Error \% | -1.52 | -0.30 | -0.62 | -1.65 | -0.61 | -0.79 | -1.96 | -1.72 | -1.57 | -1.23 |
>
> In summary, we found that the reward estimation error accumulated has a minimal effect on OPE performance measured using aggregate metrics. We also found that the error of the OPE estimation remains stable and relatively low across all target policies, including those that differ the most from the behavior policy ($\pi_0$, $\pi_9$) and result in the most significant OOD shift between target and behavior rollouts. This concludes that the prediction errors of the reward were not a significant concern in the Gym and D4RL benchmark suite. We once again thank the reviewer for raising this important point. We will clarify the aforementioned points in our revision and add the previously discussed ablation experiments to our appendix.
>
> ---
>
> ## (Weakness 1) The baselines in Figure 2 and Table 2 are inconsistent.
>
> We thank the reviewer for pointing out this typo and we will fix this in our revision.
>
> ---
>
> 1. Jiang, Nan. "A note on loss functions and error compounding in model-based reinforcement learning." arXiv, 2024.

---

> > ### Comment · Reviewer_fKxK · 2025-08-06
> >
> > Thanks for the authors' response, it has addressed most of my concerns.

---

### Official Review · Reviewer_jw9S · 2025-06-29

**Clarity:** 2
**Significance:** 3
**Originality:** 2
**Rating:** 5
**Confidence:** 4

**Summary:**

This paper proposes to address the off-policy evaluation (OPE) problem using STITCH-OPE, which leverages the denoising diffusion for long-horizon OPE in high-dimensional spaces. The authors claim two technical contributions: (1) the paper proposes a new guidance score function which can avoid over-regularization and (2) generates long trajectories by stitching partial trajectories together. The authors also provide experimental results on the standard D4RL and OpenAI Gym benchmarks and show advantages compared with existing methods on some metrics (mean squared error, correlation, and regret).

**Questions:**

The authors stress "long-horizon" settings, while I understand they are more challenging than "short-horizon" cases, but I'm trying to understand how long are the trajectories in the experiments. The authors mention "T = 768 for all D4RL problems", does that mean 768 seconds for trajectory (i.e., frequency is 1HZ), or the frequency is not 1HZ?

**Ethical Concerns:**

["NO or VERY MINOR ethics concerns only"]

**Final Justification:**

The paper studies an important problem and proposes an effective approach -- the rebuttal helps address my previous concerns.

**Limitations:**

See weaknesses and questions.

**Paper Formatting Concerns:**

None.

**Quality:**

3

**Strengths And Weaknesses:**

Strengths:

1. First, the paper studies an important problem, off-policy evaluation, which is critical in domains where deploying a new policy directly might be risky and better understanding on its impact beforehand using behavior data and policy is necessary.

2. The proposed negative behavior guidance is novel and integrating both the target and behavior policy guidance. The theoretical evaluation error upper bound result is encouraging by removing the exponential term in the previous approaches.

3. The experimental design is reasonable with convincing benchmarks and baselines. The empirical results validate the authors' hypothesis.

Weaknesses:

1. I'm not fully convinced by a claimed innovation/contribution, i.e., the "sub-trajectory stitching using a sliding window". If my understanding is correct, the authors are using a sliced history trajectory as the diffusion input and output a sliced window future trajectory. I think this is also proposed in literature, e.g., the diffusion policy paper [7]. How is the method in this paper different existing method? More clarifications would be appreciated.

2. Does the authors try different time horizon under the same benchmark? For example, in a D4RL setup, try T from 100, 200, to 700, and (I believe some readers would be curious to) see how the proposed approach work as the horizon becomes longer, compared with baselines. This can also be helpful to better validate and understand the theoretical error upper bound.

---

> ### Author Rebuttal · Authors · 2025-07-31
>
> We thank the reviewer for your insightful comments. We also agree that the problem of OPE is an important problem that has also not been fully addressed in the context of high-dimensional, long-horizon prediction. We thank the reviewer for acknowledging that our use of negative behavior guidance to address over-regularization and our OPE-specific theoretical bounds were not studied in prior diffusion work, and appreciate the feedback that our empirical results were convincing. Please find our answers to your questions below.
>
> ---
>
> ## (Weakness 1) I'm not fully convinced by a claimed innovation/contribution, i.e., the "sub-trajectory stitching using a sliding window".
>
> Thank you for pointing out the apparent similarity between STITCH-OPE and the receding-horizon approach in Diffusion Policy (cited as [7] in our paper). There are a number of key distinctions between STITCH and Diffusion Policies, most notably:
> 1. the difference in meaning of sliding windows between the two methods (action vs. state + action prediction),
> 2. the difference between training and inference procedures of the two methods, and
> 3. OPE-specific innovations in STITCH -- such as theoretical analysis showing exponential reduction in return variance when using sliding windows -- that are not discussed in the diffusion policy paper.
>
> **Distinct Objectives:** Diffusion Policy and variants apply a sliding window to predict actions for online control, thus all windows are derived from a single demonstration or from the same trajectory. In contrast, the goal of STITCH-OPE is to produce accurate synthetic trajectories (states and actions) for off-policy evaluation that remain accurate over long horizon and in the presence of distribution shift. This is achieved by drawing short segments from different parts of the offline data set, allowing the diffusion model to optimally compose/stitch them end-to-end dependent on the target policy guidance.
>
> **Distinct Training/Inference Procedures:** Diffusion Policy is designed for imitation learning by training on complete demonstration trajectories, slicing them only at inference time. Thus, there is no mechanism to recombine slices from different real trajectories. In contrast, STITCH-OPE trains the diffusion model on overlapping windows (sampled randomly) within demonstration trajectories. This allows us to build an explicit model of the length $w$ transition dynamics that can then generalize well across all possible starting states (see row B of Table 1) and even target policies (see row A of Table 1).
>
> **OPE Specific Innovations:** The unique training and inference setting in STITCH-OPE enables:
> 1. bridging states across distinct offline trajectories,
> 2. applying positive guidance to correct distribution shift across policies and even policy classes,
> 3. applying negative guidance to prevent over-regularization, and
> 4. provide a theoretical framework for analysing OPE perforamnce (under weak assumptions) that guarantees **exponential reductions in variance of the return estimate** as compared to other trajectory diffusion models.
>
> The combination of these innovations, none of which have been discussed by Diffusion Policy variants nor other prior work on trajectory diffusion, are critical to ensure reliable OPE performance across many long-horizon tasks (Figure 2, Table 2).
>
> ---
>
> ## (Weakness 2) Does the authors try different time horizon under the same benchmark?
>
> Thank you for raising this point. To address this, we ran an additional experiment to show that STITCH-OPE remains effective over a large range of evaluation horizons.
>
> Specifically, we evaluated STITCH-OPE against all baselines on the Hopper-medium data set for various values of $T \in \lbrace 128, 256, 512, 768, 1000 \rbrace.$ All values have been normalized by the global minimum and maximum achieved via the aggregation of value estimates across all the horizons. We used a discount factor of $\gamma = 0.999$ to show the difference in performance between $T = 768$ and $T = 1000$:
>
> Normalized Log RMSE
> | Method | T=128 | T=256 | T=512 | T=768 | T=1000 |
> | :--- | :--- | :--- | :--- | :--- | :--- |
> | STITCH | **-4.342 ± 0.008** | **-3.192 ± 0.003** | **-2.653 ± 0.008** | **-1.985 ± 0.010** | **-1.613 ± 0.019** |
> | MBR | -3.661 ± 0.006 | -2.977 ± 0.006 | -2.165 ± 0.009 | -1.561 ± 0.010 | -1.236 ± 0.011 |
> | IS | -3.819 ± 0.010 | -2.042 ± 0.009 | -1.194 ± 0.007 | -0.944 ± 0.003 | -0.810 ± 0.003 |
> | PGD | -3.652 ± 0.015 | -3.026 ± 0.013 | -2.301 ± 0.007 | -1.816 ± 0.006 | -1.452 ± 0.003 |
> | FQE | -2.228 ± 0.001 | -1.364 ± 0.001 | -0.991 ± 0.001 | -0.811 ± 0.002 | -0.691 ± 0.002 |
> | DR | -3.489 ± 0.024 | -2.454 ± 0.002 | -1.649 ± 0.024 | -1.317 ± 0.012 | -1.140 ± 0.018 |
>
> In summary, we found that:
> 1. STITCH-OPE performance degrades very slowly as $T$ increases, validating the theoretical bound of Theorem 3.3 empirically,
> 2. as predicted, model-based and importance sampling based methods degrade rather quickly as $T$ increases, whereas the performance of STITCH-OPE remains more stable across $T$, and
> 3. STITCH-OPE outperforms all other baselines according to MSE across all values of $T$, striking the optimal balance between fully auto-regressive and existing trajectory diffusion methods.
>
> We will include results for all metrics (rank correlation and Regret@1) for this ablation experiment in our revised version.
>
> ---
>
> ## (Question) Does that mean 768 seconds for trajectory (i.e., frequency is 1HZ), or the frequency is not 1HZ?
>
> When we write $T = 768$, we refer to the number of decision epochs in each evaluation episode. Indeed, as you correctly pointed out, it would imply 768 seconds at a frequency of 1Hz.

---

> > ### Comment · Reviewer_jw9S · 2025-08-04
> >
> > Thank you for your rebuttal. I'm convinced by the proposed framework given the additional explanation and experiments. I will raise my score.

---

### Official Review · Reviewer_8yJQ · 2025-07-02

**Clarity:** 3
**Significance:** 2
**Originality:** 3
**Rating:** 4
**Confidence:** 4

**Summary:**

STITCH-OPE introduces a model-based off-policy evaluation method for high-dimensional, long-horizon RL tasks. It leverages guided diffusion models pretrained on behavior data to generate target policy trajectories by "stitching" short sub-trajectories end-to-end. It subtracts the behavior policy’s score during guidance to mitigate distribution shift and over-regularization, and splits generating the whole long trajectories into stitching a series of short sub-trajectories. The paper provides both theoretical and experimental analysis to support the effectiveness of STITCH-OPE.

**Questions:**

1. Could you detail the specific diffusion sampling algorithm (e.g., DDPM, DDIM) and hyperparameters (e.g., diffusion steps, guidance scale, noise schedule) used for generating sub-trajectories? The paper mentions "conditional diffusion" but lacks implementation specifics critical for reproducibility.

1. The choice of $T=768$ for D4RL (vs. the standard episode length 1000) appears arbitrary. What empirical or theoretical justification supports this?

1. The paper follows the protocol described in DOPE [1] to train target policies, using SAC on D4RL medium datasets. As offline RL has advanced significantly since 2021, could STITCH-OPE's effectiveness be demonstrated using higher-performing target policies? For instance, evaluating policies trained with modern offline algorithms (e.g., CQL, IQL, TD3+BC) on "medium-replay" or "medium-expert" datasets would better assess performance on near-expert policies.

1. The model-based (MB) baseline appears outdated. Given recent progress, could you include a stronger diffusion model baseline like Diffusion World Model (DWM) [2], which also performs trajectory-level modeling suitable for OPE? Similarly, please confirm if other baselines (e.g., PGD, DRE, DR) represent the most current competitive methods or could be updated, as current baselines are generally before 2021. Related work in Section N should also be updated.

[1] Fu, Justin, et al. "Benchmarks for deep off-policy evaluation." ICLR 2021

[2] Ding, Zihan, et al. "Diffusion world model: Future modeling beyond step-by-step rollout for offline reinforcement learning." preprint 2024.

**Ethical Concerns:**

["NO or VERY MINOR ethics concerns only"]

**Final Justification:**

The rebuttal has solved most of my concerns. This paper designs a novel algorithm to the OPE problem, and the results are both significant and solid.

**Limitations:**

Yes.

**Paper Formatting Concerns:**

No.

**Quality:**

3

**Strengths And Weaknesses:**

Strengths:
- The paper is well-written and easy to follow.
- The theoretical analysis provides an exponential reduction in MSE, which allows further improvement following its route.
- It emperically ourperforms all baselines on commonly-used D4RL benchmarks and various target policies, including complex diffusion policies.

Weaknesses: see questions.

---

> ### Author Rebuttal · Authors · 2025-07-31
>
> We thank the reviewer for the detailed comments. We appreciate that the reviewer found value in our theoretical analysis, which indeed shows an exponential reduction in variance versus existing methods, and our thorough empirical evaluation and competitive OPE performance across all tasks and metrics.
>
> ---
>
> ## (Q1) Could you detail the specific diffusion sampling algorithm (e.g. DDPM, DDIM) and hyperparameters?
>
> Our implementation of the diffusion model builds on the PyTorch trajectory diffuser implementation by Janner et al. [7], which is a DDPM whose hyperparameters are described in Appendix C of that paper. We are thoroughly committed to reproducible research, and as such have provided key details regarding the diffusion model implementation in Appendix J in the supplementary material (and configuration files with detailed hyperparameters in our attached code base). We will describe all hyperparameters from [7] that we used that were omitted from Table 10 of our appendix in our revised version.
>
> ---
>
> ## (Q2) The choice of $T = 768$ for D4RL (vs. the standard episode length 1000) appears arbitrary. What empirical or theoretical justification supports this?
>
> Thank you for highlighting an important implementation detail of our work. In summary, the main reason for choosing $T = 768$ as the evaluation horizon for the generative baselines was that time steps beyond this contributed little variance to the cumulative return. We also ran an ablation over $T$ to demonstrate that STITCH-OPE works across different values of $T$.
>
> **Ablation Experiment over $T$:** Since the goal of our paper is to long-horizon correctness of OPE, we conducted an additional ablation over multiple values of $T \in \{128, 256, 512, 768, 1000\}$ on Hopper-medium plus all baselines. We also set the discount factor to $\gamma = 0.999$, to better see the performance differences across methods for $T = 1000$.
>
> Normalized Log RMSE
> | Method | T=128 | T=256 | T=512 | T=768 | T=1000 |
> | :--- | :--- | :--- | :--- | :--- | :--- |
> | STITCH | **-4.342 ± 0.008** | **-3.192 ± 0.003** | **-2.653 ± 0.008** | **-1.985 ± 0.010** | **-1.613 ± 0.019** |
> | MBR | -3.661 ± 0.006 | -2.977 ± 0.006 | -2.165 ± 0.009 | -1.561 ± 0.010 | -1.236 ± 0.011 |
> | IS | -3.819 ± 0.010 | -2.042 ± 0.009 | -1.194 ± 0.007 | -0.944 ± 0.003 | -0.810 ± 0.003 |
> | PGD | -3.652 ± 0.015 | -3.026 ± 0.013 | -2.301 ± 0.007 | -1.816 ± 0.006 | -1.452 ± 0.003 |
> | FQE | -2.228 ± 0.001 | -1.364 ± 0.001 | -0.991 ± 0.001 | -0.811 ± 0.002 | -0.691 ± 0.002 |
> | DR | -3.489 ± 0.024 | -2.454 ± 0.002 | -1.649 ± 0.024 | -1.317 ± 0.012 | -1.140 ± 0.018 |
>
> In summary:
> 1. STITCH-OPE performance degrades very slowly as $T$ increases, validating the theoretical bound of Theorem 3.3
> 2. MBR and IS methods degrade rather quickly as $T$ increases, whereas the performance of STITCH-OPE remains more stable across $T$, and
> 3. STITCH-OPE outperforms all other baselines according to MSE across all $T$, striking the optimal balance between fully auto-regressive and trajectory diffusion methods.
>
> We will include results for all metrics (rank correlation and Regret@1) for this ablation in our revised version.
>
> **Practical Justification for $T = 768$:** We agree that our choice of $T=768$ could benefit from more explicit justification.
>
> 1. An important distinction is that the truncation to $T=768$ applies **specifically to generative methods** such as STITCH-OPE and PGD, and not truncation of the data set. In contrast, **non-generative baselines** (e.g., FQE, DRE, DR, IS) use the full offline dataset without truncation, processing entire episodes up to 1000 steps -- our implementation reflects this to ensure fair comparison.
>
> 2. The primary rationale for $T=768$ in STITCH-OPE is tied to the discount factor $\gamma=0.99$, which we borrow from standard work in Offline RL and D4RL benchmarks [2]. With $\gamma=0.99$, rewards collected towards the end of the rollout contribute negligibly to the discounted return on these problems. Truncation to $T=768$ thus captures nearly all meaningful return variance across all three D4RL benchmark problems.
>
> 3. Additionally, 768 was selected as a convenient multiple for implementation efficiency, since it aligns with common batch sizes and diffusion step multiples in our codebase.
>
> ---
>
> ## (Q3) Could STITCH-OPE's effectiveness be demonstrated using higher-performing target policies?
>
> We would like to clarify a misunderstanding. Our target policies are not the SAC checkpoints from the D4RL medium datasets; they are trained online to span the full spectrum from near-random to near-optimal. Concretely, our best performing policies achieve the following raw returns, which are in par with the strongest results achieved by recent offline methods on the same tasks (medium-expert dataset).
>
> | Task     | Ours | BCQ | CQL | TD3-BC | IQL |
> |-|-|-|-|-|-|
> | HalfCheetah  | 12696 | 10875  | 9358  | 11154   | 10773  |
> | Hopper  | 2953 | 2566 | 2027  | 2900   | 3564  |
> | Walker-2d  | 4121 | 3426 | 3659  | 3806   | 3537  |
>
> Results are taken from the Robomimic documentation page [8].
>
> An added benefit of training online is that these policies are produced with fresh environment interaction (not the behaviour dataset), thus they introduce a larger shift than re‑using policies trained on the same data, making the OPE problem harder and more realistic. In summary, our current protocol already evaluates STITCH‑OPE on high‑return, high‑shift targets while remaining fully compatible with the reviewer’s request for stronger offline policies.
>
> ---
>
> ## (Q4) The model-based (MB) baseline appears outdated.
>
> We thank the reviewer for suggesting Diffusion World Models (DWM) [1] as an alternative diffusion model-based approach. We have implemented DWM in our setup and included its results in the hopper-medium benchmark below, where it is clear that STITCH-OPE (and PGD) outperform DWM.
>
> | Metric | DWM | PGD | STITCH-OPE |
> | :--- | :--- | :--- | :--- |
> | Log RMSE ↓ | -0.82 ± 0.20 | -1.22 ± 0.02 | **-2.33 ± 0.02** |
> | Rank Correlation ↑ | 0.41 ± 0.10 | 0.36 ± 0.09 | **0.76 ± 0.02** |
> | Regret @ 1 ↓ | 0.14 ± 0.11 | **0.04 ± 0.01** | 0.11 ± 0.04 |
>
> We observe that the log MSE values of DWM are worse than PGD [2], and significantly worse than STITCH-OPE which is still by far the best-performing method. This is intuitive, since DWM does not use policy guidance, and thus cannot adapt to the significant distribution shift present between the behavior and target policies.
>
> More generally, we wish to point out that while there have been significant advancements in diffusion modeling over the last few years, we believe PGD [2] is the most representative and recent model-based baseline we found to date that is **relevant for OPE**. This is because addressing distribution mismatch between behavior and target policy trajectories is a highly non-trivial problem, which only STITCH-OPE (and partially PGD) are able to tackle effectively by using positive and negative policy guidance.
>
> We will include the Diffusion World Models baseline in our evaluation (with accompanying details) and our related work section accordingly. We will also clarify why some recent diffusion work is not suited for OPE.
>
> ---
>
> ## (Q4) Please confirm if other baselines (e.g., PGD, DRE, DR) represent the most current competitive methods.
>
> We thank the reviewer for their concern over fair baselines and believe rigorous evaluation is crucial for advancing OPE. We believe our choice of baselines is up to date.
>
> **Single-Step Baselines:** Within the class of single-step dynamics modeling, the state-of-the-art in OPE still fit a single neural network to predict the next state given the current state and action, and then do Q-learning with the model rollouts [4,5,6]. While models that are one-step and auto-regressive in predicting individual state components are an alternative, Fu et al. [3] shows that using a single MLP to predict the entire next state performs better.
>
> **Trajectory Generation:** We believe that trajectory modeling is considerably more challenging in OPE, as it is largely unclear how to address the distribution shift between the behavior and the target policy. While many general diffusion modeling papers cited in our text could in principle model dynamics over multiple time steps, it is largely unclear how they can be adapted to address the distribution shift. PGD (published in 2024) [2] was the most recent paper and most representative member within this class of methods that is relevant to OPE, since it naturally corrects for the policy distribution shift using target policy guidance.
>
> **DRE, DR and Other Standard Baselines:** The other baselines in our evaluation are considered standard in the OPE literature, as evidenced by the most recent surveys on OPE [4, 5, 6]. They are also considered representative baselines across a variety of diverse classes of OPE methods, i.e. visitation densities, Q-values, doubly-robust, etc. For these reasons, we felt that it would be unfair to exclude these standard baselines, despite that some of them were introduced less recently.
>
> ---
> 1. Ding, Zihan, et al. "Diffusion world model: Future modeling beyond step-by-step rollout for offline reinforcement learning." arXiv 2024.
> 2. Jackson, Matthew T., et al. "Policy-Guided Diffusion." RLC 2024.
> 3. Fu, Justin, et al. "Benchmarks for deep off-policy evaluation." ICLR 2021.
> 4. Kiyohara, Haruka, et al. "Towards Assessing and Benchmarking Risk-Return Tradeoff of Off-Policy Evaluation." ICLR 2024.
> 5.  Uehara, Masatoshi et al. "A Review of Off-Policy Evaluation in Reinforcement Learning." arXiv 2022.
> 6. R. Figueiredo Prudencio et al., "A Survey on Offline Reinforcement Learning: Taxonomy, Review, and Open Problems," IEEE TNNLS, 2024.
> 7. Janner, Michael, et al. "Planning with diffusion for flexible behavior synthesis." arXiv 2022.
> 8. Mandlekar, Ajay, et al. "What Matters in Learning from Offline Human Demonstrations for Robot Manipulation", CoRL 2021.

---

> > ### Comment · Reviewer_8yJQ · 2025-08-04
> >
> > Thanks for your detailed response which have solved most of my concerns. I am willing to raise my rating.

---

### Official Review · Reviewer_YRRV · 2025-07-03

**Clarity:** 3
**Significance:** 3
**Originality:** 4
**Rating:** 5
**Confidence:** 3

**Summary:**

This paper presents STICH-OPE, a novel diffusion-based off-policy evaluation method for reinforcement learning. The algorithm trains a conditional diffusion model to generate short trajectories of the behavior policy. Using the flexible guidance in diffusion models, it is able to sample trajectories of the target policy nicely. The authors show that STICH-OPE performs better than various baselines through extensive experiments, and also provide a derivation of theoretical guarantees of their method.

**Questions:**

1. Clarification for guidance during the denosing process. Is the guidance applied at every step during the denoising process? The initial state action pairs are quite noisy; would they cause problems for the target and behavior policy score evaluations?
2. When target policies are diffusion policies, you mentioned that

    > Specifically, this expression cannot be calculated at $k = 0$ since $\sigma_0=0$ using the standard parameterization of diffusion models, so we approximate it at $k = 1$ and use the resulting gradient in STITCH-OPE.
    >

    Could you elaborate on what this means? Is the same gradient used throughout the whole denoising process of the STICH algorithm? Or is it just for this last denoising step, you use an approximated gradient, and all the other steps utilize the standard calculations of the score? Additionally, do these diffusion policies generate one-step actions or short action trajectories?
3. Does the trajectory dataset need to contain rollouts that are similar to the behavior policy and target policies? I’m asking this because training generative models like diffusion models is much more data-hungry than other models (like a neural dynamics model). In scenarios that require OPE (without access to simulators or when gathering new data is costly), if the training of the proposed methods requires rollouts similar to the behavior or target policy, then there’s a question of why one would still do OPE when they could already generate rollout data.
4. How sensitive is the algorithm to the selection of the behavior policy? In other words, do the return values agree when choosing different behavior policies to do the OPE process?
5. For the visualization, Figures 9 and 11. What is the optimal policy? Does it refer to the target policy being evaluated? For the OPE runs, such as STICH or PGD, are the visualizations rendered with the states’ trajectories or the action trajectories (rollout in sim)?

**Ethical Concerns:**

["NO or VERY MINOR ethics concerns only"]

**Final Justification:**

My final recommendation for this paper is 5 (Accept). I think it is a solid paper with good theoretical backups and good experiments to show the effectiveness of their methods. The rebuttal provided by the authors answered most of my questions, including several clarifications, and provided new insights on the applicability of their method on various target policies, and the target policy doesn't have to be in distribution with the training data. The missing unanswered question is the real-life application of their method. I am not too familiar with the off-policy evaluation applications except in RL algorithms. However, just from the metrics the authors provided, their STICH-OPE is very competitive against other baselines.

**Limitations:**

Yes.

**Paper Formatting Concerns:**

None.

**Quality:**

4

**Strengths And Weaknesses:**

Strengths:

1. Provided a rigorous theoretical analysis of the proposal algorithm.
2. Evaluated the proposed method on various benchmarks.
3. Overall, the paper is clear and easy to follow.

Weaknesses:

1. Most evaluations are on simple RL environments; it remains to see how the method performs in more complex and realistic RL tasks.

---

> ### Author Rebuttal · Authors · 2025-07-31
>
> We appreciate that the reviewer found our paper clear and easy to follow, our theoretical analysis rigorous and insightful, and our empirical evaluation thorough.
>
> ---
>
> ## (Q1) Is the guidance applied at every step during the denoising process?
>
> Yes, you are correct in that we apply guidance at every step of the denoising process.
>
> To clarify this, we first want to point out an interesting feature of guided diffusion not discussed in the literature. Even with a constant guidance coefficient, the actual strength of the guidance across denoising steps is varied by the denoising schedule, i.e. $\alpha_k$. In Appendix A, a concrete illustration with the standard parameterization $\sigma_k^2 = \beta_k = 1 - \alpha_k$ shows the guidance strength is $\sqrt{\alpha_k}$.
>
> **Theoretical Justification:** Although we agree that the guidance strength is larger at the initial denoising steps, we also want to clarify that this choice is necessary according to theory of classifier diffusion guidance [1]. It also parallels the standard implementation of Langevin dynamics where the ``learning rate'' is annealed over time to achieve (asymptotic) theoretical convergence guarantees.
>
> **Practical Justification:** Annealing the guidance is also intuitive, since we want to take larger steps to speed up convergence when the sample is far from its true distribution at the early denoising steps. Meanwhile, during later denoising steps, it makes practical sense to take smaller steps to ensure convergence and avoid escaping out of the high-density region. Our empirical results did not require additional tricks to improve convergence (beyond guidance normalization), and worked well across diverse target policies and policy classes.
>
> ---
>
> ## (Q2) Clarification of Diffusion Policy experiments
>
> When we refer to approximating the score at $k=1$, we refer to the calculation of the target policy's score function when the target policy is a diffusion policy. Also, our diffusion policies output immediate/single-step actions.
>
> **Details about $k = 1$:** Naturally, when the target policy is parameterized as a diffusion policy, the closed-form expression of its score function is no longer available, so we instead use the known fact that the score at denoising step $k$ is $\epsilon(a | s, k)/\sigma_k$. Since we learn $\epsilon( a|s,k)$ to match the score of the convolution of $\pi(a|s)$ with $\mathcal{N}(0, \sigma_k)$ we would ideally estimate the score function at the last denoising step $k = 0$. But this is practically impossible since $\sigma_0 = 0$ and so we use the estimate at $k = 1$. Most notably, these calculations are described in terms of the denoising process of the target policy, and not the guided denoising process of STITCH (for sampling trajectories) which is a separate process. Thus, once we derive the approximate score function of the diffusion policy, we use this score function to guide across all denoising steps within STITCH.
>
> **$k = 1$ as a Policy Approximation:** As we show in Table 3, STITCH-OPE still provides competitive OPE performance when the target policy score function must be approximated as discussed above, which introduces some measurement errors into the guidance function (which is often the case in many real-world settings). It also outperforms baselines (such as MBR or DRE) that use the target policy only for sampling actions during OPE, which are technically unbiased samples from the exact target policy.
>
> **Output of the Diffusion Policy:** All diffusion policies we use output single-step (i.e. immediate) actions. This follows exactly the offline diffusion policy literature [2], where single-step actions were also used in their practical implementation.
>
> ---
>
> ## (Q3) Does the trajectory dataset need to contain rollouts that are similar to the behavior policy and target policies? I'm asking this because training generative models like diffusion models is much more data-hungry than other models.
>
> We would like to thank the reviewer for raising this important point. We want to clarify that STITCH-OPE does **not** require that rollouts be similar between behavior and target policies. This is due to the following reasons:
>
> **Chunked Diffusion Training:** One of the key technical novelties of STITCH-OPE is training on short overlapping windows sampled from the behavior data. By reducing the prediction horizon, we cut down on the diffusion model's training complexity and data requirements compared to training on full rollouts. More importantly, by observing the same state-action dynamics across different rollouts, they can be reused at different epochs, across multiple trajectories, and across target policies. This means that the behavior data set does not need to contain rollouts that are similar to the target policies, as long as the support of the target policy's immediate action distribution is within the support of the behavior policy -- a standard assumption in OPE literature.
>
> **Compositional Inference:** At inference time, STITCH-OPE stitches together generated trajectory chunks to reconstruct full-horizon rollouts. Together with policy guidance, this process generates novel rollouts that were never observed in the behavior data. Crucially, these rollouts remain faithful to the target policy even when the target policy's action distribution is significantly different from the behavior policy, as long as the target policy stays within the support of the behavior policy. Please see row A of Table 1 for a visual example and Appendix C for a formal proof that the entropy of stitched trajectories is higher than the behavior policy.
>
> **Empirical performance:** We validate these contributions on D4RL and demonstrate that STITCH can reliably predict target policy returns even when the target policy differs from the behavior policy (low MSE, high correlation and low regret aggregated across all target policies in Table 2). We also want to clarify that the construction of the target policies in our paper of varying ability from random to expert, as well as the inclusion of target diffusion policies in our work, makes for a considerably more challenging evaluation benchmark than prior OPE work.
>
> ---
>
> ## (Q4) How sensitive is the algorithm to the selection of the behavior policy?
>
> We found that STITCH-OPE is robust to the selection of the behavior policy. To check this, we ran STITCH-OPE and baselines on the medium-expert and expert data sets from the Hopper D4RL benchmark using default parameters ($T = 768, \gamma = 0.99$). The metrics are shown below:
>
> Medium-Expert
> | Metric | FQE | DR | IS | DRE | MB | PGD | Ours (STITCH) |
> | :--- | :---: | :---: | :---: | :---: | :---: | :---: | :---: |
> | Log RMSE ↓ | -1.140 ± 0.027 | -0.709 ± 0.051 | -0.151 ± 0.002 | -0.182 ± 0.001 | -1.334 ± 0.008 | -0.554 ± 0.016 | **-2.174 ± 0.012** |
> | Rank Corr. ↑ | 0.511 ± 0.086| 0.224 ± 0.146 | 0.105 ± 0.062 | 0.132 ± 0.144 | 0.753 ± 0.011 | 0.774 ± 0.044 | **0.826 ± 0.003** |
> | Regret@1 ↓ | 0.023 ± 0.001| 0.064 ± 0.041 | 0.599 ± 0.150 | 0.131 ± 0.045 | 0.000 ± 0.000 | 0.009 ± 0.008  | **0.000 ± 0.000** |
>
> Expert
> | Metric | FQE | DR | IS | DRE | MB | PGD | Ours (STITCH) |
> | :--- | :---: | :---: | :---: | :---: | :---: | :---: | :---: |
> | Log RMSE ↓ | -0.391 ± 0.003 | 0.016 ± 0.001 | -1.054 ± 0.006 | 0.016 ± 0.001 | -1.455 ± 0.008| -0.010 ± 0.001 | **-2.328 ± 0.012** |
> | Rank Corr. ↑ | 0.46 ± 0.027 | 0.202 ± 0.113 | 0.261 ± 0.099 | 0.202 ± 0.113 | 0.277 ± 0.004  | 0.473 ± 0.029 | **0.507 ± 0.039** |
> | Regret@1 ↓ | 0.115 ± 0.042 | 0.095 ± 0.037 | 0.321 ± 0.067 | 0.095 ± 0.037 | 0.175 ± 0.001 | **0.000 ± 0.000** | 0.035 ± 0.010 |
>
> STITCH-OPE outperforms all other baselines across all three metrics using both medium-expert and expert data sets, and performance remains stable across both data sets. This is surprising, given that we would expect the distribution shift for the expert data set to be greater than the medium data set (which contains more diverse rollouts) or the medium-expert data set (which is a mixture of the prior two). This suggests that STITCH-OPE remains effective at composing trajectories across a variety of behavior data sets generated by different behavior policies. Finally, we observe a steep decline in performance of the baselines on the medium-expert and expert data sets as compared to the medium data set (Table 2 in the paper), which suggests that prior OPE work is not robust to large distribution shifts between the behavior and target policies.
>
> ---
>
> ## (Q5) For the visualization, Figures 9 and 11, what is the optimal policy?
>
> The random and optimal policies correspond to the worst and best target policies within the 10 target policies evaluated. We selected the best and worst target policies to demonstrate that STITCH-OPE can generate accurate synthetic trajectories even when guided by target policies that are significantly different from each other and from the behavior policy. The trajectories labeled as ``STITCH Guided`` or ``PGD Guided`` are purely synthetic trajectories generated from the trained diffusion model with policy guidance, i.e. there is no simulator involved.
>
> ---
>
> 1. Dhariwal, Prafulla et al. "Diffusion models beat gans on image synthesis." NeurIPS, 2021.
> 2. Wang, Zhendong, et al. "Diffusion Policies as an Expressive Policy Class for Offline Reinforcement Learning." ICLR, 2023.

---

> > ### Comment · Reviewer_YRRV · 2025-08-05
> >
> > Thank you for your rebuttal. It has addressed most of my questions. I will maintain my score.

---

### Note · Authors · 2025-08-13

We thank the AC and reviewers for the constructive discussion. We are grateful that our rebuttal was well-received.

The novel contributions of STITCH-OPE include sub-trajectory stitching with conditional diffusion and negative behavior guidance for off-policy evaluation in RL. Our analysis shows an exponential MSE reduction in window length $w$ (vs. horizon $T$), and experiments on D4RL/Gym show consistent gains in MSE, rank correlation, and regret. Our work takes a crucial step in addressing the longstanding challenges of OPE by providing a rigorous framework for long-horizon evaluation, a critical gap in the literature.

To further strengthen the paper and directly address the reviewers' excellent suggestions, we conducted four new experiments during the rebuttal period:

- **Horizon Ablation:** We validated STITCH-OPE's robustness across various long horizons.
- **Modern Baseline Comparison:** We implemented and compared against the suggested Diffusion World Models (DWM) baseline, demonstrating STITCH-OPE's superior performance.
- **OOD Robustness:** We conducted an ablation on learned vs. true rewards to empirically address concerns about out-of-distribution performance, showing minimal impact.
- **Behavior Policy Sensitivity:** We evaluated our method on additional D4RL datasets (medium-expert, expert) to confirm its robustness to the choice of behavior data.

We are also grateful for the reviewers' additional feedback and we will update our paper based on that. Specifically:
- **Enhanced Literature Review:** Update model-based OPE section to better contextualize recent diffusion-based approaches and explain why some methods aren't suitable for OPE.
- **Technical Clarifications:** Add detailed explanations of guidance application during denoising and diffusion policy score approximation methods.
- **Data Requirements Discussion:** Clarify how chunked diffusion training reduces data requirements compared to full-trajectory methods.

We believe these new results, which we will integrate into the camera-ready version, significantly bolster our paper's claims.

Thank you for your time and consideration.

Sincerely,

Authors

---

### Decision · Program_Chairs · 2025-09-17

**Decision:**

Accept (spotlight)

**Comment:**

This paper introduces a diffusion framework for off-policy evaluation that stitches shorter sub-trajectories to construct long trajectories and uses a negative behavior guidance mechanism to address distribution shift.

All of the reviewers were positive about the paper for its technical soundness, theoretical analysis, and strong empirical performance on benchmarks. While there were initial questions regarding the novelty of the stitching component, the choice of baselines, and specific hyperparameters, the authors provided a comprehensive and convincing rebuttal, successfully addressing all concerns.

Given the novelty, strong theoretical and empirical backing, and I believe that this work is significant enough to gather interests from the diffusion & reinforcement learning community.